# Evidence for control of cerebral neurovascular function by circulating platelets in healthy older adults

Gabriella M. K. Rossetti[1,2] , Joanne L. Dunster[3], Aamir Sohail[1,4], Brendan Williams[1] , Kiera M. Cox[3], Suzannah Rawlings[3], Elysia Jewett[1,3], Eleanor Benford[1,3] , Julie A. Lovegrove[3,5] , Jonathan M. Gibbins[3] and Anastasia Christakou[1]

[1]*Centre for Integrative Neuroscience and Neurodynamics (CINN), School of Psychology and Clinical Language Sciences, University of Reading, Reading, UK*
[2]*Institute of Sport, Department of Sport and Exercise Sciences, Manchester Metropolitan University, Manchester, UK*
[3]*Institute for Cardiovascular and Metabolic Research (ICMR), School of Biological Sciences, University of Reading, Reading, UK*
[4]*Centre for Human Brain Health (CHBH), School of Psychology, University of Birmingham, Birmingham, UK*
[5]*Institute of Food and Nutritional Health (IFNH), Department of Food and Nutritional Sciences, University of Reading, Reading, UK*

Handling Editors: Harold Schultz & Vaughan Macefield

The peer review history is available in the Supporting Information section of this article (https://doi.org/10.1113/JP288405#support-information-section).

This article was first published as a preprint. Rossetti GMK, Dunster JL, Sohail A, Williams B, Cox KM, Jewett E, Benford E, Lovegrove JA, Gibbins JM, Christakou A. 2024. Evidence for direct control of neurovascular function by circulating platelets in healthy older adults. bioRxiv. https://doi.org/10.1101/2024.05.31.596788

The Journal of Physiology

**Abstract figure legend** Epidemiological evidence suggests a link between dementia and cardiovascular health, including the regulation of platelets. To investigate this relationship, we examined how platelet reactivity relates to cerebral neurovascular function and systemic vascular reactivity in healthy older adults. Fifty-two participants were recruited, and platelet reactivity was assessed using plate-based aggregometry (dose-response to platelet agonists and inhibitors). Cerebral neurovascular coupling was measured using functional magnetic resonance imaging (fMRI) during cognitive tasks, whereas vascular reactivity was assessed using vasoactive agents [ADP, sodium nitroprusside (SNP) and $CO_2$]. The data show that elevated platelet reactivity is associated with blunted (delayed, shorter and smaller) cerebral blood flow responses to neuronal activation. Importantly, this association was not explained by systemic vascular reactivity, suggesting the mechanism extends beyond systemic atherosclerosis.

**Abstract** Platelets play a vital role in preventing haemorrhage through haemostasis, but complications arise when platelets become overly reactive, leading to pathophysiology such as atherothrombosis. Elevated haemostatic markers are linked to dementia and predict its onset in long-term studies. Despite epidemiological evidence, the mechanism linking haemostasis with early brain pathophysiology remains unclear. Here, we aimed to determine whether a mechanistic association exists between platelet function and cerebral neurovascular function in 52 healthy mid- to older-age adults. To do this, we combined, for the first time, magnetic resonance imaging of cerebral neurovascular function, peripheral vascular physiology and *in vitro* platelet assaying. We show an association between platelet reactivity and cerebral neurovascular function that is both independent of vascular reactivity and mechanistically specific: Distinct platelet signalling mechanisms (ADP, collagen-related peptide, thrombin receptor activator peptide 6) were associated with different physiological components of the haemodynamic response to neuronal (visual) stimulation (full-width half-maximum, time to peak, area under the curve), an association that was not mediated by peripheral vascular effects. This finding challenges the previous belief that systemic vascular health determines the vascular component of cerebral neurovascular function, highlighting a specific link between circulating platelets and the neurovascular unit. Because altered cerebral neurovascular function marks the initial stages of neurodegenerative pathophysiology, understanding this novel association is now imperative, with the potential to lead to a significant advancement in our comprehension of early dementia pathophysiology.

(Received 7 January 2025; accepted after revision 10 April 2025; first published online 28 May 2025)

**Corresponding author** G. M. K. Rossetti: Institute of Sport, Department of Sport and Exercise Sciences, Manchester Metropolitan University, 99 Oxford Road, Manchester, M1 7EL United Kingdom, UK.    Email: g.rossetti@mmu.ac.uk

## Key points

- Haemostasis (platelet function) has been linked to the early stages of dementia, but the precise mechanisms are not well understood.
- This study considers whether a causal mechanism exists through atherothrombotic effects on the vasculature which can in turn affect brain health, or through platelet-specific interactions with brain physiology.
- Here, we show that elevated platelet reactivity is associated with blunted (delayed, shorter and smaller) cerebral blood flow responses to neuronal activation in healthy middle-aged and older adults.
- However, the association between platelet reactivity and cerebral neurovascular function was not mediated by systemic vascular reactivity.
- This finding challenges the previous belief that systemic vascular health determines the vascular component of cerebral neurovascular function, highlighting a specific link between circulating platelets and the neurovascular unit in early dementia pathophysiology.

## Introduction

Dementia is a major cause of disability and dependency among the elderly, affecting over 55 million people worldwide and with an annual economic cost of 1.3 trillion US dollars (World Health Organization, 2018). Clinical evidence suggests a link between dementia and cardiovascular health, including the regulation of platelets. Although platelets play a vital role in our physiological well-being by preventing haemorrhage through haemostasis (blood clotting), pathological conditions arise when platelets become overly reactive, leading to complications such as atherothrombosis (Davì & Patrono, 2007). Understanding the potential implications of platelet function for dementia may elucidate preventive measures and therapeutic interventions aimed at addressing this pressing public health concern.

Haemostatic markers including platelet reactivity and pro-clotting factor concentrations (e.g. von Willebrand factor), are elevated in individuals with vascular dementia (Quinn et al., 2011). This association between platelet function and dementia extends beyond vascular dementia, as demonstrated in the Framingham Heart Study, where platelet aggregation response to adenosine diphosphate (ADP) at middle-age independently predicted both all-cause dementia at 20-year follow-up (Ramos-Cejudo et al., 2022). Indeed, platelet reactivity outweighed other commonly acknowledged risk factors, including education, low-density lipoprotein-cholesterol and body mass index (BMI). On the basis of such associations, the use of anti-thrombotic medications has been proposed as a potential approach to slow the progression of the disease in individuals with dementia (Cortes-Canteli et al., 2019; Grammas & Martinez, 2014). However, the underlying physiological mechanisms require further elucidation before appropriate interventions can be effectively implemented (Sun & Langer, 2022). Taken together, this suggests that targeted prophylactic intervention on platelet function holds promise for prolonging neurocognitive health in ageing. However, evidence is limited to clinical dementia states where it is impossible to establish mechanistic links or differentiate cause and consequence.

A possible explanation of the observed relationship between platelet reactivity and dementia may be through systemic vascular effects. Indeed, dysregulation of the haemostatic system, leading to unwarranted platelet activation and subsequent thrombosis, forms an early and central aspect of cardiovascular disease pathophysiology (Renga & Scavizzi, 2017). At the same time, subclinical vascular dysregulation emerges as an early pathological event in the development of dementia, manifesting years before detectable amyloid beta or tau abnormalities (Iturria-Medina et al., 2016). Furthermore, cardiovascular and metabolic health conditions, including hypertension (Ramos-Cejudo et al., 2022), heart disease (Abete et al., 2014) and diabetes (Gudala et al., 2013), are risk factors for mild cognitive impairment, vascular dementia and Alzheimer's disease (Tarumi & Zhang, 2018).

Alternatively, platelets may directly interact with the 'neurovascular unit', a complex network of neurons, glial cells, and blood vessels that regulate cerebral blood flow in response to neural activity. Supporting this, high plasma fibrinogen levels (a marker of haemostasis) have been associated with a progressive decline in cognitive abilities among the elderly, even after accounting for cardiovascular morbidity and associated risk factors (Rafnsson et al., 2007), suggesting a potential link between platelet activity and brain function.

Notably, the signalling mechanisms involved in the regulation of platelets share similarities with those governing vascular tone. For example, nitric oxide (NO) and prostanoids are vasodilators that are essential for synaptically-driven initiation of the haemodynamic response (Iadecola, 2017), and also act as potent inhibitors of platelet aggregation (Gkaliagkousi & Ferro, 2011; Jin et al., 2005). Considering these interconnected factors, platelets may provide a valuable blood biomarker of cardiovascular-associated risk for dementia, facilitating the development of targeted and effective prevention or treatment strategies.

Despite growing evidence for associations between platelet function and dementia, the current evidence is dependent on disease state comparisons (i.e. dementia *vs.* healthy), behavioural cognitive effects (e.g. non-verbal reasoning in Rafnsson et al., 2007) and beta-amyloid deposition. These disease processes occur later than the dysregulation of blood vessels (Iturria-Medina et al., 2016; Sweeney et al., 2018) and potentially much later than neurovascular dysfunction, which is proposed to be

**Gabriella Rossetti** is a Lecturer in Human Neuroscience in the Department of Sport and Exercise Sciences at Manchester Metropolitan University. She conducted this research while working as a Postdoctoral Research Fellow at the Centre for Integrative Neuroscience and Neurodynamics at the University of Reading. Now in an academic position, she is expanding on these findings by exploring the role of platelets across the timeline of neurodegenerative pathophysiology and their clinical applications. Her research focuses on the intricate relationship between physiology and cognitive function, with a particular interest in neurovascular coupling and region-specific responses to systemic physiological stressors.

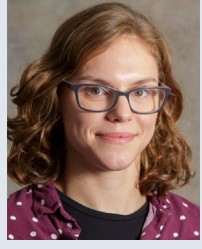

the first point of failure in the pathophysiological cascade (Girouard & Iadecola, 2006; Iadecola, 2017). Despite its importance, assessment of cerebral neurovascular function in humans is complex and thus rarely performed. However, a growing number of studies have demonstrated consistent effects by estimating the model parameters of the haemodynamic response function (HRF) using blood oxygen level dependent (BOLD) functional magnetic resonance imaging (fMRI). Specifically, older age (West et al., 2019), diabetes (Guimarães et al., 2024; van Duarte et al., 2015) and subjective cognitive decline (Lu et al., 2022) are associated with altered haemodynamic responses to neural activity that are smaller in magnitude (peak height), delayed (longer time to peak; TTP), and of shorter duration (smaller full-width half-maximum; FWHM) compared to controls. These reduced haemodynamic responses probably reflect problems matching the vascular response to the energy demand in the active brain region. The unexplored inter-actions between platelets and cerebral neurovascular function could be crucial for developing targeted pre-ventive strategies against dementia, potentially trans-forming global health outcomes.

We therefore sought to investigate whether a relationship exists between platelet reactivity and cerebral neurovascular function in the general population of middle- to older-aged healthy adults. Platelet reactivity was assessed using platelet reactivity multiparameter phenotyping analysis and cerebral neurovascular function was assessed by estimating haemodynamic responses (i.e. HRFs) to a fixed visual stimulus. We incorporated experimental manipulations that targeted peripheral vascular reactivity and cerebrovascular reactivity to determine whether any association could be attributed to systemic vascular effects, or interactions between platelets and the neurovascular unit.

## Methods

### Ethical approval

A favourable opinion for conduct was obtained from the University of Reading Research Ethics Committee (ref: UREC 21/11, accepted 14/06/2021) and the study was conducted following the standards of the World Medical Association *Declaration of Helsinki* 2013 (World Medical Association, 2013), with written informed consent obtained from all participants.

### Study design

The study consisted of cross-sectional data collection that occurred over two study visits, separated by 7–31 days (mean $\pm$ SD: 17.7 $\pm$ 8.9 days). The *a priori* objectives of

the study were: (1) to determine whether platelet reactivity was associated with aspects of cerebral neurovascular function; (2) to investigate whether specific aspects of platelet reactivity were associated with cerebral neuro-vascular function (mechanistic selectivity); and (3) to test whether any association between platelet reactivity and cerebral neurovascular function could be explained by the state of the peripheral and cerebral vasculature. After initiation of the data analyses, we additionally sought (4) to determine whether correlations between platelet reactivity and cerebral neurovascular function were attributable to platelet reactivity *per se*, or simply reflected shared variance driven by demographic differences.

Fifty-two healthy middle-aged and older adults (50–80 years; 28 females) were recruited into the study (mean $\pm$ SD: aged 63.6 $\pm$ 7.7 years; BMI 25.5 $\pm$ 4.1 kg m$^{-2}$). Eleven participants were excluded from the final analysis sample because of insufficient platelet or fMRI data for the primary analyses (final $N = 40$; 22 females; aged 61.9 $\pm$ 6.5 years; BMI 25.4 $\pm$ 4.1 kg m$^{-2}$). Where other individual subjects were excluded from specific analyses, this is highlighted in the analysis description. Participants had no cardio-vascular disease, diabetes, liver disease, hypertension, bleeding disorder, neuropsychiatric condition, current psychopathology or sleep disturbance. Participants were not taking anti-coagulant, anti-platelet or psychoactive medications. Education level was 5 $\pm$ 2 across the sample and ranged from level 2 (e.g. GCSE grade A* to C) to level 8 (e.g. PhD).

### Experimental procedures

For both visits, participants consumed a standardised meal the night before and fasted for 12 h before each study visit. Participants avoided strenuous exercise and alcohol and were advised to get a good night's sleep the day before the visit. All study visits started at 08.00 h. Visit 1 included some procedures and measures not reported in this publication but described in brief here for accuracy. Visit 1 consisted of a 29 min encoding session for a sub-sequent memory task and a clinical assessment of memory function (not included in this publication), followed by a blood sample for platelet reactivity assaying, and MRI of anatomical measures and functional MRI to measure task-evoked haemodynamic (i.e. BOLD) responses in the primary visual cortex (V1) during visual stimulation (and in multiple regions during an episodic memory task; not reported here). Visit 2 consisted of laser doppler imaging (LDI) with iontophoresis of vasoactive agents to assess peripheral vascular reactivity followed by MR Imaging using arterial spin labelling (ASL) to measure cerebral blood flow (CBF) (reliability across visit 1 and visit 2: intraclass correlation coefficient = 0.76; $R = 0.77$;

$P < 0.001$), hypercapnic cerebrovascular reactivity (5% $CO_2$) and hypocapnic CVR (hyperventilation).

### Haemostatic function: platelet reactivity

**Sample collection.** Blood samples were taken via venepuncture and collected into vacutainers containing 3.8% (w/v) sodium citrate. Platelet-rich plasma (PRP) was prepared by centrifuging whole blood at 100 $\textbf{\textit{g}}$ for 20 min and used within 30 min.

**Plate-based aggregometry assay.** End-point aggregation measurements were taken using a plate-based assay in PRP using 96-well half-area assay plates (Greiner, Kremsmünster, Austria) pre-prepared with freeze dried platelet agonists [collagen-related peptide (CRP), ADP and thrombin receptor activator peptide 6 (TRAP-6)] over a range of concentrations to cover the full concentration dose response for each, as described in Dunster et al. (2021). PRP was first treated with vehicle, PAPA-NONOate (100 μM) or iloprost (1.25 nM) and incubated for 10 min at 37°C before the transfer of 40 μL to the assay plate. Plates were shaken at 1200 rpm for 5 min at 30°C using a plate shaker (Quantifoil Instruments, Grosslöbichau, Germany) and absorption of 405 nm light measured using a FlexStation (Molecular Devices, San Jose, CA, USA).

**Assessment of platelet reactivity and multicomponent phenotyping.** Data were analysed using the in-house R package PPAnalysis to extract the metrics sensitivity ($EC_{50}$) and capacity ($E_{max}$) and enable multiparameter phenotyping. Principal component analysis was conducted on the vehicle data (platelet reactivity to activating agonists) using the prcomp function from the R package stats and clustered using an agglomerative hierarchical algorithm, as per Dunster et al. (2021). This confirmed our previously published finding that platelet sensitivity and platelet capacity are distinct (Fig. 1*B*; see also Appendix, Fig. A1). This analysis provided individual assessment of latent platelet reactivity components for each participant and phenotype stratification of platelet reactivity across the sample (Fig. 1). All platelet phenotypes were found to be stable assessed using the R package pvclust (Suzuki & Shimodaira, 2006) with 1000 bootstrap iterations (Group 1, bp = 97%; Group 2, bp = 98%; Group 3, bp = 98%). Individual assay conditions (including competing platelet activating and inhibiting manipulations) were then interrogated to determine whether specific aspects of platelet reactivity were associated with cerebral neurovascular function.

### Cerebral neurovascular function

**Visual stimulation task.** A black-and-white, radial checkerboard stimulus was presented to the participants to induce activation in the primary visual cortex (V1). The polarity of the checkerboard was reversed at a frequency of 8 Hz (every 125 ms), which has been shown to robustly activate V1 in older adults (Uchiyama et al., 2021) and has been used in previous studies investigating cerebral neurovascular coupling in V1 (Leontiev & Buxton, 2007). The checkerboard stimuli were presented in 6.8 s blocks, with jittered inter-trial intervals (ITIs) between each period of visual stimulation which were optimised using optseq, version 2.0 (Centre for Functional Neuroimaging Technologies, Massachusetts General Hospital, Harvard, MA, USA). During the ITIs, a blank grey screen with a white fixation cross in the centre was displayed on the monitor (this was also presented for 2.72 s at the beginning of the task and 6.8 s at the end of the task). The total duration of the task was 310 s (5 min and 10 s). The task was programmed using the psychtoolbox-3 extension of MATLAB (MathWorks, Natick, MA, USA) (Brainard, 1997; Kleiner et al., 2007; Pelli, 1997) and was presented to the participants on a monitor with a resolution of $1024 \times 768$ and a refresh rate of 60 Hz.

**MRI acquisition and analysis.** MRI data were collected on a Prisma-fit 3T scanner using a 32-channel head coil (Siemens, Munich, Germany).

**Anatomical reference: T1 MP-RAGE.** High resolution T1-weighted images were acquired as an MP-RAGE sequence: time of echo (TE) = 2.29 ms, time of repetition (TR) = 2300 ms, time of inversion (TI) = 900 ms, flip angle (FA) = 8°, field of view (FOV) = 240 × 240 × 180 mm, voxel dimensions = $0.9 \times 0.9 \times 0.9$ mm$^3$, 192 slices (slice oversampling 16.7%), acquisition time = 5 min 21 s. T1 images were acquired for each scanning session and used for registration of the CBF and fMRI scans to the MNI 152 2mm brain template using FMRIB's Linear Image Registration Tool (FLIRT, FSL) (Jenkinson et al., 2002).

**Functional data: fMRI BOLD.** To measure task-evoked activations, BOLD contrast functional images were acquired using an echo-planar (EPI) sequence (TR = 1360 ms, TE = 30 ms, FA = 90°, FOV = 192 × 192 × 100 mm$^3$, voxel dimensions = $3 \times 3 \times 4$ mm$^3$, 25 slices). Echo-planar data quality were assessed for qualitative artefact (signal distortion, e.g. ringing) and quantitative excessive motion (relative or absolute movement >2 mm) by agreement of two researchers assessing independently (GMKR and AS). The data were pre-processed in FSL including brain extraction (BET2), slice timing correction

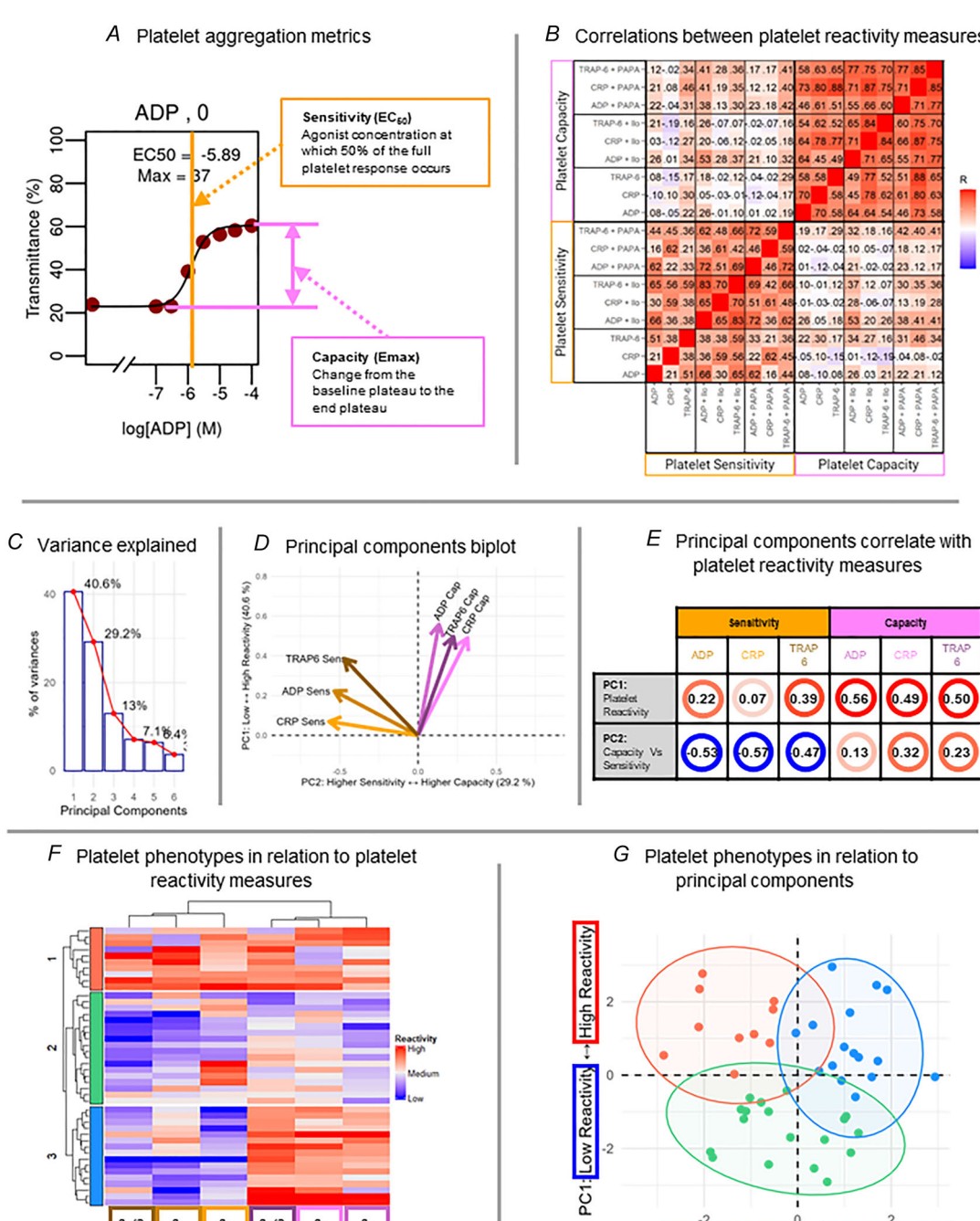

**Figure 1. Platelet reactivity methods: multiparameter phenotyping**

Platelet reactivity was assessed by a plate-based aggregometry (PBA) assay. For illustrative purposes, sensitivity variables are highlighted in orange, and capacity variables are highlighted in pink throughout the panels. Platelet reactivity (sensitivity and capacity) was tested in response to agonists adenosine diphosphate (ADP), collagen-related peptide (CRP) and thrombin receptor activator peptide 6 (TRAP-6). Reactivity was measured to each agonist alone, and in the combined presence of each agonist with the following inhibitors: PAPA-NONOate (PAPA; a nitric oxide donor) and Iloprost (Ilo; synthetic prostacyclin). *A*, concentration dose-response curves were analysed to extract the metrics sensitivity ($EC_{50}$) and capacity ($E_{max}$). *B*, correlations between platelet reactivity measures confirmed sensitivity and capacity are distinct concepts. Values displayed are Spearman's rho, with colour reflecting the strength of the relationship. Latent platelet reactivity characteristics were identified using principal component analysis (PCA) on the agonist-only conditions (without the presence of PAPA or Ilo inhibition). *C*, the first

two principal components accounted for most of the variance (cumulative variance = 69.8%) and were interpreted using (*D*) PCA biplot and (*E*) patterns of correlations with the original platelet reactivity measures. Agglomerative hierarchical clustering identified three distinct platelet phenotype groups (G1, *N* = 10; G2, *N* = 18; G3, *N* = 16) which were reviewed in respect to (*F*) the original platelet reactivity measures and (*G*) the principal components from the PCA. [Colour figure can be viewed at wileyonlinelibrary.com]

(interleaved), motion correction (MCFLIRT), spatial smoothing (FWHM = 5 mm), high-pass filtering, B0 unwarping (fieldmap TR = 0.488 ms, TE1 = 0.00519 ms, TE2 = 0.00765, FA = 60°) and registration (FLIRT).

**Estimation of HRFs.** We calculated individual HRFs from pre-processed echo-planar data using a Finite Impulse Response model in FSL with ten, 2 s impulses as regressors (onset from the start of each visual stimulation block, with a combined duration of 20 s). Individual statistical parametric maps for each impulse were then concatenated for each individual participant to create 4D niftis comprising of voxel-wise beta weights across the whole brain with each volume representing a 2 s impulse. A V1 region of interest (ROI) mask from the Harvard–Oxford subcortical atlas (Desikan et al., 2006) (thresholded at 50% and binarised) was transformed into subject space for each participant, and used to extract the HRF time series for the V1 ROI of the visual stimulus regressor, creating a HRF that was specific to the participant, regressor, and ROI. HRF time series were then up-sampled to a frequency of 0.2 s and smoothed using a spline regression (R package = zoo (Zeileis et al., 2023)).

HRF parameters were calculated as described below (Fig. 2):

- FWHM (s) = width of the impulse at half of the peak height.
- Area under the curve (AUC) = total quantity encompassed by the HRF impulse
- TTP (s) = time at which the peak height occurs.
- Peak height (%) = maximum BOLD signal value across the HRF impulse.

Each individual HRF was qualitatively assessed by an experienced neuroscientist (GMKR) to determine whether it described physiologically-plausible task-evoked haemodynamics. HRF time series that were not physiologically plausible, or where the HRF could not be confidently identified, were excluded on the basis any extracted HRF parameters would reflect only noise. This approach was necessary for the broader study, which included cognitive brain regions beyond the visual cortex, where the robustness of the HRF varied. HRF parameters were calculated systematically but constrained by local minima corresponding to qualitative assessment of a single HRF impulse (see Appendix, Fig. A2). This was necessary because the duration covered by HRF time series was set at 20 s across all cases, but clearly covered multiple HRF impulses in some cases.

Experimental manipulation check: visual stimulation evoked individualised HRFs

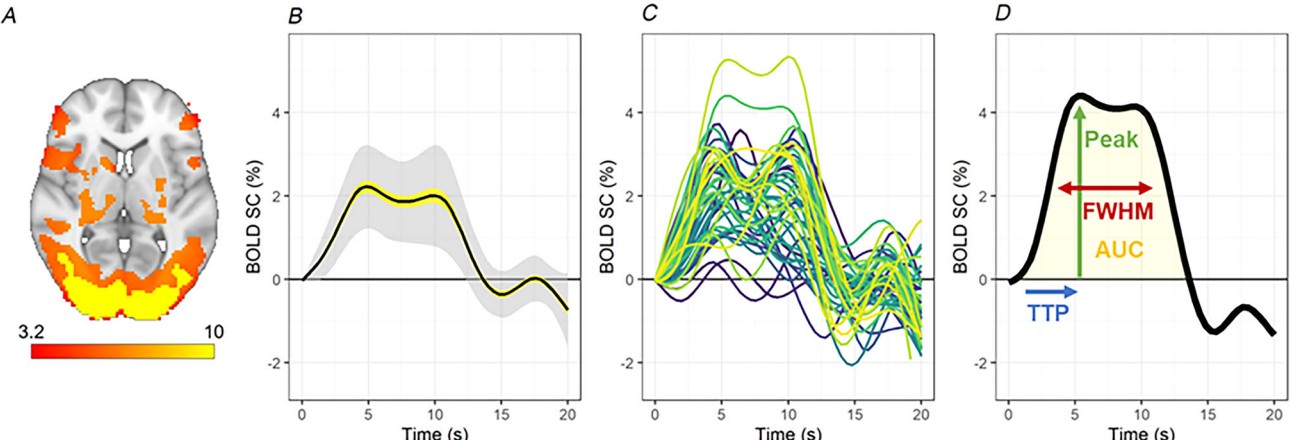

**Figure 2. Visual stimulation evoked individualised haemodynamic response functions (HRFs)**
Visual stimulation using flashing checkerboard was confirmed to activate (trigger a blood flow response) in the primary visual cortex (V1). This is demonstrated by (*A*) Group-level activation map (colour = *z* score); (*B*) group-level average (± SE in yellow shading, ± SD in light grey shading) haemodynamic response function (HRF) and (*C*) subject-specific individual HRFs (colours differentiate participants). *D*, subject-specific individual HRFs were then characterised by the parameters; full-width half-maximum (FWHM), area under the curve (AUC), time to peak (TTP) and peak height (peak). [Colour figure can be viewed at wileyonlinelibrary.com]

An experimental manipulation check was conducted to ensure the visual stimulation task selectively activated the a priori ROI. Specifically, the visual stimulus regressor was associated with activations in V1. Group-level activation maps utilising convolution with optimal basis functions (FMRIB's Linear Optimal Basis Sets, FLOBS) (Woolrich et al., 2004) demonstrated clear activation localised to the visual cortex (Fig. 2*A*) and recognisable HRFs were present in V1 during visual stimulation, but not in control regions (see Appendix, Fig. A3). A large degree of individual variability existed regarding the shape of the HRF, confirming the premise that task-evoked haemodynamics are not uniform, and may provide valuable information regarding individual cerebral neurovascular function. Individual parameters were then extracted from each HRF (Fig. 2).

Of note, BOLD HRF parameters were not correlated with each other. This suggests that, at least in the context of short-duration stimulation (6.8 s) with a reversing visual checkerboard, the different HRF parameters capture different components of cerebral neurovascular function. Specifically, the BOLD FWHM did not correlate with TTP ($R = 0.062$; $P = 0.7$) or peak height ($R = 0.17$; $P = 0.28$) and the TTP did not correlate with peak height ($R = 0.097$; $P = 0.53$).

### Systemic vascular function

The maximum change in perfusion in response to a vaso-active stimulus is termed vascular reactivity and is an important biomarker to assess vascular function. In the present study, vascular reactivity to vasoactive stimuli in the peripheral vasculature (arm) and cerebrovasculature (brain) was assessed to determine whether any associated between platelet reactivity and cerebral neurovascular function was explained by systemic vascular reactivity or cerebrovascular reactivity to vasoactive agents.

**Peripheral vascular reactivity to NO.** Both endothelium-dependent (ACh) and endothelium independent (sodium nitroprusside, SNP) peripheral microvascular reactivity were assessed using LDI (moorLDI2; Moor Instruments, Axminster, UK) with iontophoresis. Simultaneous delivery of ACh (Sigma-Aldrich, Merck Ltd, Darmstadt, Germany) and SNP (Sigma-Aldrich, Merck Ltd) was performed using an iontophoresis controller (MIC2; Moor Instruments) to assess endothelium-dependent and endothelium independent cutaneous perfusion, respectively. Perfusion changes in response to the delivery of both vasoactive drugs were assessed on the participant's volar aspect of the right forearm. Following two basal measurements of skin perfusion, an incremental constant current was delivered using the LDI software. Current delivery was

progressively increased in 5 mA steps (5, 10, 15 and 20 mA) to yield a total charge of 8 mC within the first 12 min, with five additional scans at the end delivering no current (0 μA) and therefore without the delivery of the vasoactive agents. In total, 21 scans were performed. ACh and SNP were diluted to 1% solutions with 0.5% saline and delivered simultaneously into the skin via anode (ACh) and cathode (SNP) internal electrode Perspex chambers (diameter 22 mm) (ION 6; Moor Instruments). The scans were performed simultaneously with the iontophoresis protocol. The scan protocol lasted 21 min and all scans were performed with ambient lighting restricted.

Measurements of perfusion were conducted offline using the moorLDI Review V6.1 software. Perfusion values were quantified for ACh and SNP calculating the median for each ROI. Results are presented as the percentage change in perfusion from the baseline scan collected immediately before the drug delivery, and was calculated as:

$$\text{Change in perfusion (\%AU)}$$
$$= \frac{\text{Peak perfusion (AU)} - \text{Baseline perfusion (AU)}}{\text{Baseline perfusion (AU)}}$$
$$\times 100$$

**Cerebrovascular reactivity to $CO_2$.** To measure cerebrovascular function, CBF measurements were acquired (1) at rest; (2) in response to hypercapnia (5% $CO_2$); and (3) in response to hyperventilation-induced hypocapnia (30 breaths $min^{-1}$) (Fujii et al., 2014). Both hypercapnic CVR (CVRhyper) and hypocapnic CVR (CVRhypo) were calculated relative to individual change in end-tidal carbon dioxide ($P_{ETCO_2}$).

For all conditions, CBF was measured using ASL. Images were acquired using the vessel-encoded PICORE Q2T pseudo-continuous ASL (pCASL) package. pCASL data are collected using an EPI readout (TR = 4000 ms, TE = 13.0 ms, TI = 1800 ms, FA = 90°, FOV = 220 × 220 × 118 $mm^3$, voxel size = 3.4 × 3.4 × 4.5 $mm^3$, 24 axial slices with 10% slice gap). Labelling of inflowing blood was achieved through a parallel slab applied below the acquisition slices (bolus duration = 700 ms, post-labelling delay (PLD) = 1800 ms post-labelling delay, 76 volumes, duration = 5 min and 6 s). Interleaved tag and control pairs were acquired with a tag RF FA of 20° and a duration of 600 μs, with 1000 μs separation. Perfusion and arrival time were quantified using the BASIL toolbox (Chappell et al., 2009) within FSL (version 6.0.1).

*Hypercapnic cerebrovascular reactivity (CVRhyper)*: Hypercapnic CVR (CVRhyper) was calculated relative to

individual change in end-tidal carbon dioxide ($P_{ET}CO_2$), using:

$$CVR_{hyper} = \frac{\%\Delta CBF_{hyper}}{\Delta P_{ETCO_{2_{hyper}}}}$$

where

$$\%\Delta CBF_{hyper} = \frac{CBF_{hyper} - CBF_{baseline}}{CBF_{baseline}} \times 100$$

and

$$\Delta P_{ETCO_{2_{hyper}}} = P_{ETCO_{2_{hyper}}} - P_{ETCO_{2_{baseline}}}$$

*Hypocapnic cerebrovascular reactivity (CVRhypo)*: To assess the decrease in vasodilatation in response to a vasoconstricting stimulus, participants hyperventilated to induce hypocapnia for 5 min, whereas a 5 min pcASL scan was acquired using the same parameters as for the baseline rCBF measurement detailed above. Hyperventilation of 30 breaths $min^{-1}$ was achieved through participants breathing in time to a visual cue. Based on previous literature, this degree of hyperventilation should induce $\sim$20 mmHg decrease in end-tidal $CO_2$. Hypocapnic CVR ($CVR_{hypo}$) was calculated using:

$$CVR_{hypo} = \frac{\%\Delta CBF_{hypo}}{\Delta P_{ETCO_{2_{hypo}}}}$$

where

$$\%\Delta CBF_{hypo} = \frac{CBF_{hypo} - CBF_{baseline}}{CBF_{baseline}} \times 100$$

and

$$\Delta P_{ETCO_{2_{hypo}}} = P_{ETCO_{2_{hypo}}} - P_{ETCO_{2_{baseline}}}$$

### Statistical analysis

All statistical analyses were conducted in R, version 4.2.2 (R Foundation, Vienna, Austria). To determine whether platelet reactivity was associated with aspects of cerebral neurovascular function, Spearman's rho correlations were conducted between principal component individual weightings and HRF parameters [FWHM (s); AUC; TTP (s); peak height (%), peak]. A $3 \times 1$ (phenotype group $\times$ HRF parameter) ANOVA was then conducted for each HRF parameter to compare the parameter between platelet phenotype groups. To investigate whether specific aspects of platelet reactivity (mechanistic selectivity) were associated with cerebral neurovascular function platelet reactivity, Spearman's rho correlations were conducted between individual platelet assay conditions and HRF parameters. Correction for multiple comparisons by false discovery rate (FDR) was conducted across all correlations in the study, including those not reported in this article (FDR corrected alpha = 0.05). To determine whether

correlations between platelet reactivity and cerebral neurovascular function simply reflected shared variance driven by demographic differences, partial correlations were run accounting for demographic variables of age, sex and BMI. Finally, serial mediation models were conducted using the lavaan package (Rosseel, 2012) to test whether the association between platelet reactivity (sensitivity to ADP) and cerebral neurovascular function (HRF FWHM) was explained by a mechanistic pathway through peripheral and cerebral vascular function. The total indirect effect (a1*d21*b2) was estimated through the model design, whereas the significance of individual mediator indirect effects was calculated using the distribution of the product of coefficients (Sobel test).

## Results

### Platelet reactivity was associated with specific components of cerebral neurovascular function

**Platelet reactivity multiparameter phenotyping revealed associations between platelet reactivity and cerebral neurovascular function.** Platelet reactivity phenotyping facilitates the investigation of the underlying causes and consequences of variations in platelet function, thereby advancing progress towards precision medicine (Dunster et al., 2021). In the present study, we used multiparameter phenotyping to estimate latent platelet reactivity components for each participant aiming to stratify the sample (Fig. 1) and investigate association with aspects of cerebral neurovascular function.

The latent platelet reactivity components and phenotype groups were then used to investigate the effect of platelet reactivity on cerebral neurovascular function assessed by estimating local changes in blood flow in response to neural activity in the primary visual cortex (V1). The model parameters were then extracted from individual participant HRFs (Fig. 2). Specifically, we calculated the FWHM, which reflects the duration of the neurovascular response; TTP, which reflects the speed of the initial neurovascular response; AUC, which reflects the total size of the HRF; and peak height (peak), which reflects the maximum magnitude of the neurovascular response.

Platelet reactivity was associated with the duration (HRF FWHM) and size (HRF AUC) of the blood flow response in the V1. Specifically, higher platelet reactivity (low $\rightarrow$ high PC1, 40.6% variance in platelet responses) (Fig. 1) was associated with shorter FWHM ($R = -0.45$, $P = 0.006$) (Fig. 3A) and smaller AUC ($R = -0.41$, $P = 0.015$) (Fig. 3B).

Participants were then grouped into subsets (phenotypes) characterised by similar patterns in platelet responses. Clustering the data revealed distinct and robust platelet phenotype groups (Fig. 1F and G, see Methods:

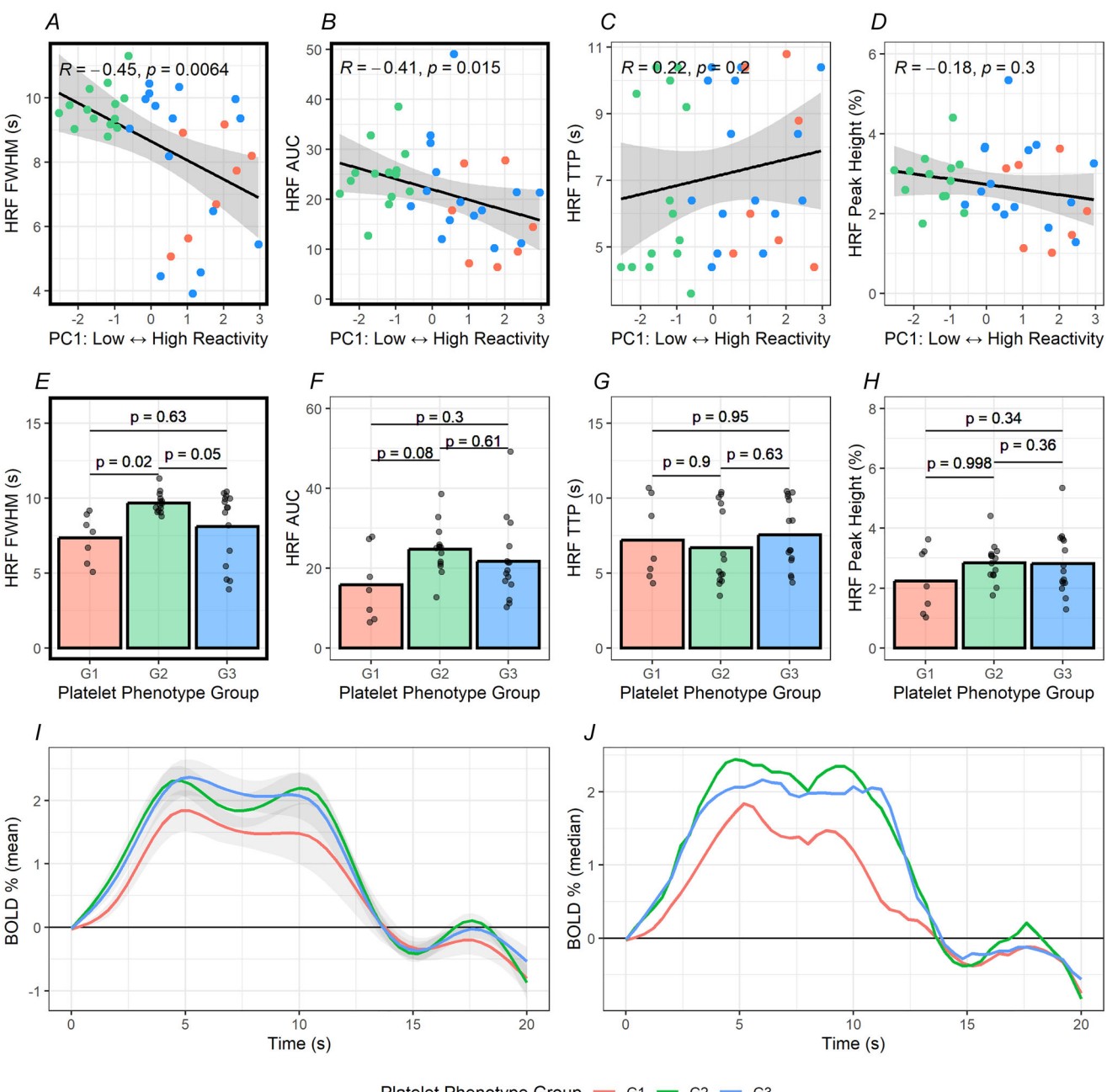

**Figure 3. Platelet reactivity multiparameter phenotyping revealed associations between platelet reactivity and cerebral neurovascular function**

For all panels colour indicates platelet phenotype group [red (G1), $N = 10$; green (G2), $N = 18$; blue (G3), $N = 16$]. Higher platelet reactivity was associated with (*A*) shorter neurovascular response in the primary visual cortex (V1) measured by haemodynamic response function (HRF) full-width half-maximum (FWHM) and (*B*) smaller neurovascular response measured by HRF area under the curve (AUC). Platelet reactivity was not associated with (*C*) the speed of the initial neurovascular response (time to peak; TTP) or (*D*) the maximum magnitude of the neurovascular response (peak height). *E*, one of the high platelet reactivity groups (Group 1, see Fig. 1*D* and *E*) had a shorter FWHM compared to the low reactivity group (Group 2). There were no differences between platelet phenotype groups in (*F*) AUC), (*G*) TTP or (*H*) peak height. The (*I*) mean ± SD and (*J*) median haemodynamic response function (HRF) values for each platelet phenotype. [Colour figure can be viewed at wileyonlinelibrary.com]

Assessment of platelet reactivity and multicomponent phenotyping). These platelet phenotype groups were able to differentiate cerebral neurovascular function even in this healthy population. Specifically, the duration of the neurovascular response in the V1 (HRF FWHM) was significantly different between the platelet phenotype groups (main effect $F = 5.032$, $P = 0.01$, $\eta^2 = 0.236$) (Fig. 3*E*), with pairwise group differences revealing a shorter FWHM in one of the high platelet reactivity groups (Group 1) compared to the low reactivity group [Group 2; $\Delta$–2.34, (–0.34 to –4.34), $P = 0.02$] (Fig. 3*E*). There were no significant differences between any other platelet phenotype groups [Group 1 – Group 3, $\Delta$–0.75, (–2.73 to 1.23), $P = 0.63$; Group 2 – Group 3; $\Delta$1.59, (–0.02 to 3.19), $P = 0.05$].

Multiparameter phenotyping did not reveal any associations between platelet reactivity and the speed of the initial neurovascular response in the V1 (HRF TTP) or the maximum magnitude of the neurovascular response (peak height). Specifically, platelet reactivity (low → high, PC1) (Fig. 1*C–E*) was not associated with TTP ($R = 0.21$, $P = 0.20$) (Fig. 3*C*) or peak height ($R = -0.21$, $P = 0.20$) (Fig. 3*D*) of the HRF. Furthermore, there were no differences between platelet phenotype groups in AUC ($F = 2.566$, $P = 0.09$, $\eta^2 = 0.135$) (Fig. 3*F*), TTP ($F = 0.476$, $P = 0.63$, $\eta^2 = 0.041$) (Fig. 3*G*) or peak height ($F = 2.364$, $P = 0.11$, $\eta^2 = 0.064$) (Fig. 3*H*).

The second principal component contributing to platelet reactivity (higher sensitivity → higher capacity, PC2, 29.2% variance in platelet responses) (Fig. 1*C–E*) was not associated with any measure of cerebral neurovascular function (FWHM, $R = 0.26$, $P = 0.13$; AUC, $R = 0.12$, $P = 0.49$; TTP, $R = 0.29$, $P = 0.08$; peak height, $R = 0.12$, $P = 0.47$).

Taken together, this establishes that higher platelet reactivity is associated with a shorter (HRF FWHM) and smaller (HRF AUC) CBF response to activation of the visual cortex, with platelet reactivity multiparameter phenotyping revealing discernible impacts of platelet function on cerebral neurovascular function within healthy middle-age and older adults.

**Evidence for mechanistic selectivity in the association between platelet reactivity and cerebral neurovascular function.** Platelet regulation is complex and involves a balance between multiple activation and inhibition regulatory mechanisms, some of which overlap with signalling mechanisms governing cerebral neurovascular function. In particular, NO and prostanoids are key inhibitory regulators of platelet aggregation (Gkaliagkousi & Ferro, 2011; Jin Ma et al., 2005) and are simultaneously essential components of cerebral neurovascular function governing haemodynamic responses to neural activity (Iadecola, 2017). To determine whether specific aspects of platelet reactivity were associated with cerebral neurovascular function (HRF parameters), we analysed individual platelet assay conditions with competing platelet activating and inhibiting manipulations (Fig. 4; see also Appendix, Fig. A4). This provided evidence to suggest that there may be mechanistic selectivity that connects platelet reactivity and cerebral neurovascular function.

Specifically, higher platelet sensitivity ($EC_{50}$) and capacity ($E_{max}$) were associated with shorter duration of the neurovascular response (HRF FWHM) (Fig. 4*A* and *B*). This effect was observed most strongly when platelet aggregation was triggered by ADP but was also present when platelet aggregation was triggered by TRAP-6. Platelet reactivity (sensitivity or capacity) was not associated with FWHM when platelet aggregation was triggered by CRP. This association between platelet reactivity and HRF FWHM was specific to the agonist used to trigger aggregation but was not affected by the presence or absence of competing platelet inhibition.

Furthermore, higher platelet sensitivity (but not capacity) was associated with an overall smaller neurovascular response (HRF AUC), but only when platelet aggregation was triggered by TRAP-6 (Fig. 4*C*). This association was specific to platelet reactivity to TRAP-6 without competing inhibition from iloprost (a synthetic prostanoid) or PAPA-NONOate (an NO donor).

Lastly, higher platelet capacity (but not sensitivity) was associated with a slower initiation of the neurovascular response (longer HRF TTP), but only when platelet aggregation was triggered by CRP (Fig. 4*F*). This association was specific to platelet responses to CRP and persisted with the addition of competing inhibition by iloprost (a synthetic prostanoid). By contrast, the addition of PAPA-NONOate (an NO donor) removed the association between platelet capacity and HRF TTP.

No measure of platelet reactivity appeared to correlate with the maximum magnitude of the neurovascular response (HRF peak height) (Fig. 4*G* and *H*).

Combined, the above findings provide evidence for mechanistic selectivity, such that specific platelet signalling mechanisms are associated with different physiological components of the haemodynamic response to neural activation. Although the functional implications of the observed effects are not fully understood, they are consistent with those observed in ageing (West et al., 2019), diabetes (Guimarães et al., 2024; van Duarte et al., 2015) and subjective cognitive decline (Lu et al., 2022), and are indicative of age-related attenuation in matching blood flow to energy demand in the active brain region.

**The association between platelet reactivity and cerebral neurovascular function is not attributable to demographic differences.** Partial correlations

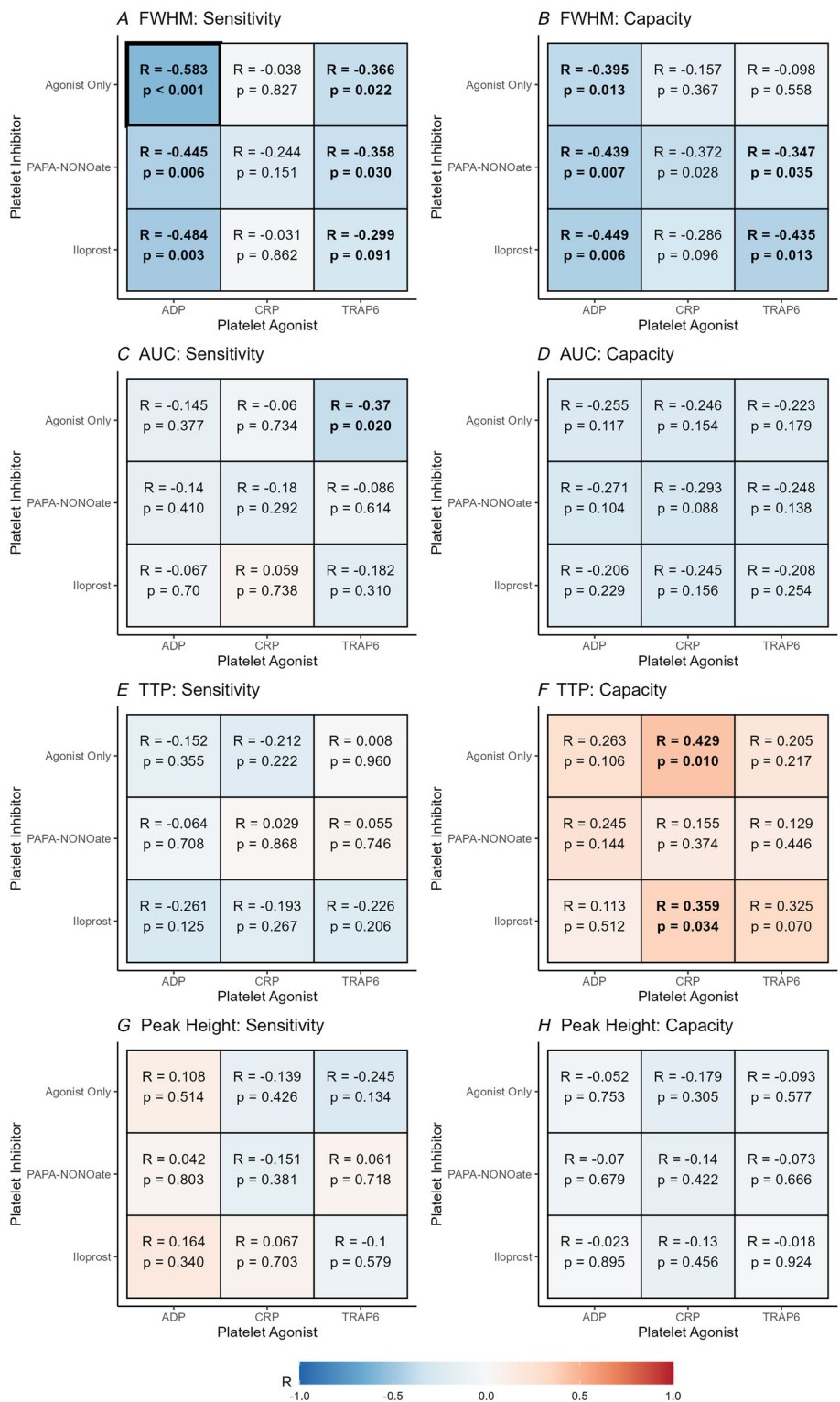

**Figure 4. Evidence for mechanistic selectivity in the relationship between platelet reactivity and cerebral neurovascular function**

*A*, relationships between platelet sensitivity ($EC_{50}$) and HRF full-width half-maximum (FWHM). *B*, relationships between platelet capacity ($E_{max}$) and HRF FWHM. *C*, relationships between platelet sensitivity and HRF area under the curve (AUC). *D*, relationships between platelet capacity and HRF AUC. *E*, relationships between platelet sensitivity and HRF time to peak (TTP). *F*, relationships between platelet capacity and HRF TTP. *G*, relationships between platelet sensitivity and HRF peak height. *H*, relationships between platelet capacity and HRF peak height.

All correlations are Spearman's rho due to non-parametric data. Significant correlations ($P < 0.05$) are in bold text, whereas correlations surviving multiple comparison correction (false discovery rate; FDR) are displayed with bold borders. Scatter plots of each correlation are presented in the Appendix (Fig. A4). [Colour figure can be viewed at wileyonlinelibrary.com]

accounting for demographic variables of age, sex and BMI were conducted to determine whether the observed associations between platelet reactivity and cerebral neurovascular function simply reflected shared variance because of demographics (Supplementary Table and Fig. 5). Alternatively, additional relationships between platelet reactivity and cerebral neurovascular function could be obscured because of demographic effects.

Our analyses suggested no direct influence of age or sex on cerebral neurovascular function (no significant partial correlations). Higher BMI was associated with smaller AUC and smaller peak height, suggesting high BMI is independently associated with smaller blood flow responses to neuronal activity (Supplementary Table).

Importantly, correlations between platelet reactivity and cerebral neurovascular function accounting for demographic variables were not meaningfully different from the total correlations (Fig. 5). This suggests that the association between platelet reactivity and cerebral neurovascular function does not simply reflect shared variance driven by demographic differences. In particular, the relationships between the multiparameter phenotyping-derived measure of latent platelet reactivity with shorter FWHM (total correlation $R = -0.45$, $P = 0.006$; partial correlation $R = -0.46$, $P = 0.007$) and smaller AUC (total correlation $R = -0.41$, $P = 0.015$; partial correlation $R = -0.55$, $P < 0.001$) were not meaningfully changed when accounting for age, BMI and sex. Platelet sensitivity (ADP and TRAP-6) and capacity (ADP) remained significantly correlated with HRF FWHM, platelet sensitivity (TRAP-6) remained significantly correlated with HRF AUC and platelet capacity (CRP) remained significantly correlated with HRF TTP. Two additional correlations became significant: (1) platelet capacity with TRAP-6 + iloprost was significantly associated with longer HRF TTP after accounting for age, sex and BMI (total correlation $R = 0.33$, $P = 0.070$; partial correlation $R = 0.41$, $P = 0.026$) and (2) platelet sensitivity to TRAP-6 was significantly associated with smaller HRF peak height after accounting for age, sex and BMI (total correlation $R = -0.25$, $P = 0.134$; partial correlation $R = -0.37$, $P = 0.027$).

Therefore, although our data provides some indication that sex and BMI may affect cerebral neurovascular function as assessed by HRF parameter estimates, our observed associations between platelet reactivity and cerebral neurovascular function are not attributable to demographic factors.

**Correction for multiple comparisons.** The data included in this publication are part of a larger study that incorporates BOLD HRF parameters from other brain regions (outside V1; posterior cingulate cortex, precuneus, motor cortex) and cognitive tasks (episodic memory) obtained in the same scanning sessions. We chose to take the conservative approach to correct for multiple comparisons over all BOLD HRF parameters, platelet variables, and cognitive tasks included in the study (a total of 288 comparisons), including those not described in this publication.

Only one correlation remained after correction by FDR to an alpha of 0.05 (Fig. 4*A*, indicated by black box outline). Platelet sensitivity to ADP (alone, without inhibitor) was negatively correlated with FWHM in V1 during visual stimulation ($R = -0.58$, FDR adjusted $P = 0.028$). This correlation also remained after FDR correction when accounting for demographic variables of age, sex and BMI ($R = -0.59$, $P = 0.0001$, FDR adjusted $P = 0.029$). Notably, ADP is the only agonist from the assay that not only triggers and potentiates aggregation, but also is produced by platelets *in vivo* (Gachet, 2006). This autocrine activation loop amplifies platelet aggregation and thrombus formation, underscoring the importance of platelet ADP responses in vascular physiology and pathology.

The correlation between platelet sensitivity to ADP (alone, without inhibitor) and HRF FWHM was then interrogated to determine whether it was explained by systemic vascular function (combined peripheral and cerebral) or either peripheral vascular function or cerebrovascular function alone.

## Platelet reactivity affects cerebral neurovascular function, independent of vascular reactivity

**Systemic vascular function does not explain the association between platelet reactivity and cerebral neurovascular function.** Given that dysregulation of the haemostatic system is a central aspect of cardiovascular disease pathophysiology (Renga & Scavizzi, 2017) and, concurrently, vascular dysregulation is a key contributor to dementia pathophysiology (Iturria-Medina et al., 2016), we assessed whether the association between platelet reactivity and cerebral neurovascular function was explained through systemic vascular function. We used a serial mediation model to test whether the association between platelet reactivity (sensitivity to ADP) and cerebral neurovascular function (HRF FWHM) was

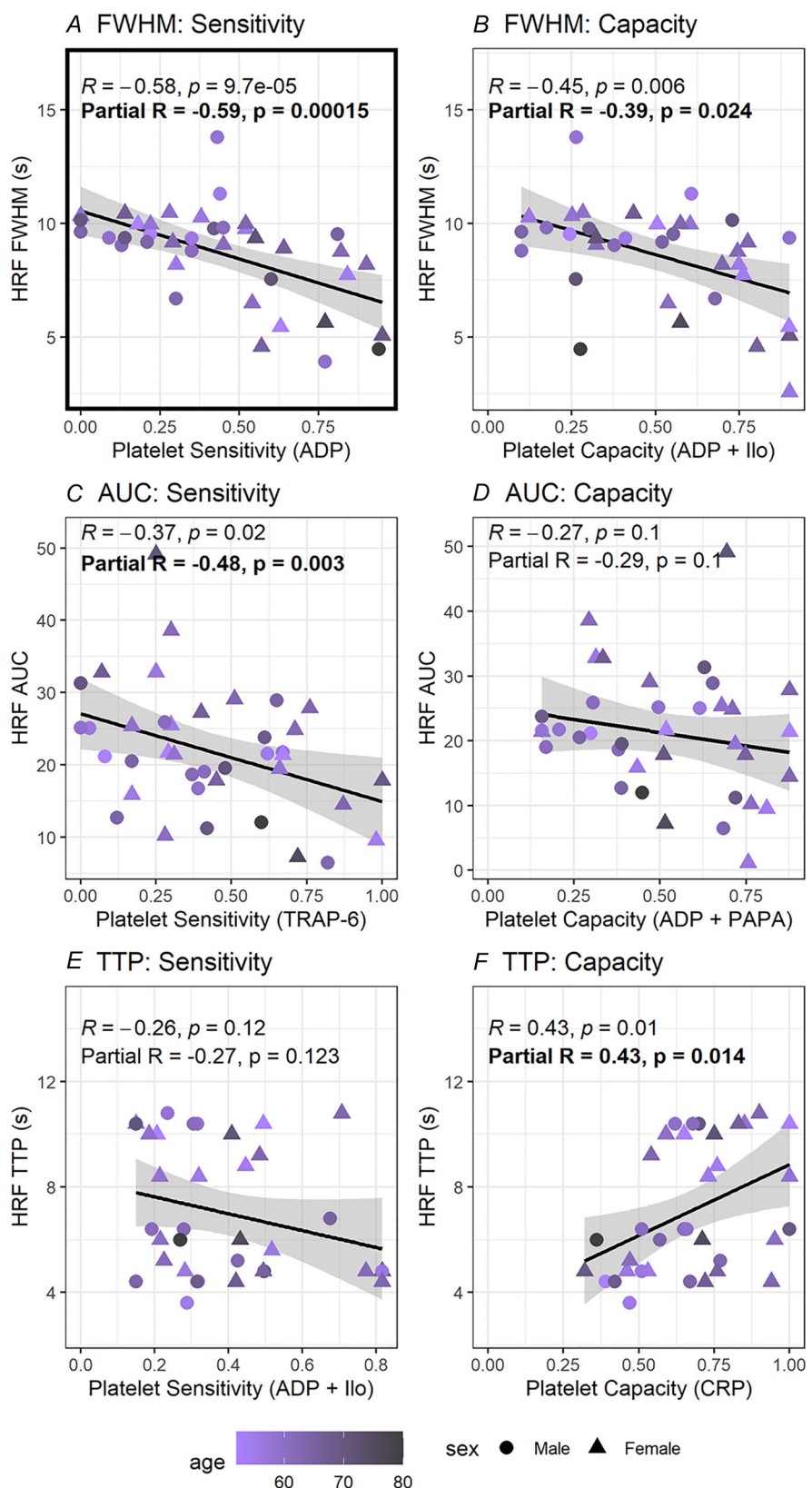

**Figure 5. Relationships between platelet reactivity and cerebral neurovascular function are not attributable to demographic differences**
The individual strongest correlation from each panel of Fig. 4 is included here for illustrative purposes. Original correlations displayed in Fig. 4 were re-run accounting for demographic variables of age (colour), sex (shape) and

BMI (not shown on figure). Partial correlations between platelet reactivity and cerebral neurovascular function (haemodynamic response function (HRF) parameters) accounting for demographic variables (partial *R*) were not meaningfully different from the total correlations (*R*). *A*, relationship between platelet sensitivity to ADP (agonist only) and HRF full-width half-maximum (FWHM). *B*, relationship between platelet capacity with ADP (+ iloprost) and HRF FWHM. *C*, relationship between platelet sensitivity to thrombin receptor activator peptide 6 (TRAP-6) and HRF area under the curve (AUC). *D*, relationship between platelet capacity with ADP (+ PAPA-NONOate) and HRF AUC. *E*, relationship between platelet sensitivity to ADP (+ iloprost) and HRF time to peak (TTP). *F*, relationship between platelet capacity with CRP (agonist only) and HRF TTP. [Colour figure can be viewed at wileyonlinelibrary.com]

**Table 1. Schematic representation of serial mediation analyses including mediating variables.**

| Platelet reactivity X | Peripheral vascular M1 | Cerebrovascular M2 | Neurovascular Y |
|---|---|---|---|
| Platelet sensitivity to ADP | Change in perfusion % (ACh) Change in perfusion % (SNP) | Perfusion (whole brain) – normocapnia Arrival time (whole brain) – normocapnia Perfusion (whole brain) – hypercapnia Arrival time (whole brain) – hypercapnia Perfusion (whole brain) - hypocapnia Arrival time (whole brain) – hypocapnia CVRhyper (whole brain) CVRhyper (V1) CVRhypo (V1) | BOLD FWHM in V1 (visual stimulation) |

In total, 18 serial mediation models were tested to investigate whether the association surviving correction by false discovery rate (FDR) (between platelet sensitivity to ADP and HRF FWHM) was explained by a serial pathway through peripheral vascular effects (M1) and cerebrovascular effects (M2). BOLD, blood oxygen level dependent MRI signal; CVRhyper, hypercapnic cerebrovascular reactivity; CVRhypo, hypocapnic cerebrovascular reactivity; FWHM, full-width half-maximum; SNP, sodium nitroprusside (nitric oxide donor); V1, primary visual cortex.

explained by a mechanistic pathway through peripheral (arm) and cerebral (brain) vascular function (schematic shown in Table 1). Peripheral vascular function was assessed by peripheral vascular reactivity to NO, including change in perfusion in response to Ach, which triggers NO release from the endothelium (endothelium-dependent), and in response to SNP, which is an NO donor that acts directly on the smooth muscle (endothelium independent).

Cerebrovascular function was assessed by cerebrovascular reactivity to $CO_2$, including whole brain perfusion and arrival time under different $CO_2$ conditions, and change in perfusion in response to hypercapnia (CVRhyper) and hypocapnia (CVRhypo) from the whole brain and specifically from the primary visual cortex (V1).

**Systemic vascular function: combined peripheral and cerebrovascular reactivity.** The serial mediation model revealed that the association between platelet reactivity and cerebral neurovascular function was not explained by systemic vascular function determined by combined peripheral vascular reactivity and cerebrovascular reactivity. This suggests that platelets may directly interact with cerebral neurovascular function.

No combination of peripheral and cerebral vascular function variables significantly explained the relationship between platelet sensitivity to ADP and HRF FWHM

(Table 2). Specifically, estimates of the indirect (mediating) effect ranged from –0.001 to 0.019 (*P* values in the range 0.39 to 0.99) for all combinations of the vascular function variables.

Of note, peripheral vascular reactivity was not associated with cerebrovascular reactivity, and the estimate of the M1 → M2 effect ranged from –0.17 to 0.19 (*P* = 0.25 to 0.99).

**Peripheral vascular reactivity to NO.** The mediation analysis was also used to determine whether the association between platelet reactivity and cerebral neurovascular function (pathway C) was explained specifically by peripheral vascular reactivity to NO (excluding cerebrovascular reactivity). Platelet sensitivity to ADP was associated with peripheral vascular reactivity to NO (pathway A1), but peripheral vascular reactivity was not associated with HRF FWHM (pathway B1), meaning that peripheral vascular reactivity alone did not explain the association between platelet sensitivity to ADP and HRF FWHM (Table 2).

Within the mediation model, the A1 and B1 pathways indicate whether the association between platelet sensitivity to ADP and HRF FWHM was explained by peripheral vascular reactivity. Platelet sensitivity to ADP had a moderate negative effect on both endothelium-dependent peripheral vascular reactivity (ACh) and endothelium independent peripheral vascular reactivity (SNP) (A1 path estimates ranged from –0.35 to –0.30; *P* = 0.01 to 0.04). This negative association probably reflects shared biochemical regulation of platelet inhibition and peripheral vascular tone regulation (Gkaliagkousi & Ferro, 2011; Jin Ma et al., 2005). However, peripheral vascular reactivity did not affect HRF FWHM in 17 out of the 18 mediation analyses (B1 path estimates ranged from 0.15 to 0.27, *P* = 0.04 to 0.23). Combining these two paths (A1 → B1) confirmed that the association between platelet sensitivity to ADP and HRF FWHM was not explained by peripheral vascular reactivity (indirect effect estimates ranged from –0.09 to –0.05, *P* = 0.12 to 0.30).

**Cerebrovascular reactivity to carbon dioxide CO₂.** Finally, the mediation analysis was used to determine whether the association between platelet reactivity and cerebral neurovascular function (pathway C) was explained specifically by cerebrovascular reactivity to $CO_2$ (excluding peripheral vascular reactivity). Although some measures of cerebrovascular function were associated with HRF FWHM (pathway B2), this was separate from the association with platelet sensitivity to ADP (pathway C), meaning cerebrovascular function alone did not explain the association between platelet sensitivity to ADP and visual FWHM (Table 2).

The A2 and B2 pathways of the mediation model indicate whether the association between platelet sensitivity to ADP and HRF FWHM was explained by cerebrovascular function. Platelet sensitivity to ADP had no effect on cerebrovascular function in 17 out of 18 mediation analyses (A2 path estimates ranged from –0.35 to 0.30; *P* = 0.04 to 0.96). However, some cerebrovascular function variables were associated with HRF FWHM. CBF perfusion (normocapnia) was positively associated with HRF FWHM (B2 path estimate 0.28, *P* = 0.02), CBF perfusion (hypocapnia) was positively associated with HRF FWHM (B2 path ranged from 0.28 to 0.30, *P* = 0.02) and finally, CVRhyper in the V1 was negatively associated with HRF FWHM (B2 path estimates ranged from –0.36 to 0.35, *P* = 0.002). Combining these two paths (A2 → B2) confirmed that the association between platelet sensitivity to ADP and HRF FWHM was not explained by cerebrovascular reactivity (indirect effect estimates ranged from –0.09 to 0.04, *P* = 0.11 to 0.30).

## Discussion

This is the first study to establish a mechanistic link between platelet reactivity and cerebral neurovascular function *in vivo* in humans. Higher platelet reactivity was associated with a shorter duration and overall smaller haemodynamic response to activation of the primary visual cortex. This association captures a specific link from platelet function to cerebral neurovascular coupling, one that bypasses systemic vascular function effects.

This study captured individual differences in cerebral neurovascular function in a sample of healthy middle-aged and older adults from the general population. Individuals with higher platelet reactivity exhibited attenuated (shorter, smaller and delayed) haemodynamic responses to visual cortex activation, indicative of problems recruiting the vascular response to match the energy demand in active brain regions that may in-turn contribute to neurodegeneration. The haemodynamic response is pivotal for ensuring the delivery of energy substrates and removal of waste products from the brain (Schaeffer & Iadecola, 2021; Sweeney et al., 2019). Failure to supply sufficient oxygenated blood can lead to hypoxic injury, potentially causing damage to neurons, impairing synaptic function, and ultimately resulting in brain atrophy (Kisler et al., 2017; Zlokovic, 2005). Chronic hypoxia and impaired clearance of metabolic waste products could induce neurotoxicity, triggering inflammation, oxidative stress, and causing damage to both neurons and glial cells (Iadecola, 2017). These neurotoxic processes significantly contribute to neurodegeneration (Persidsky et al., 2006). Chronically attenuated haemodynamic responses may initiate a

**Table 2. Systemic vascular function does not explain the relationship between platelets and cerebral neurovascular function: serial mediation analyses.**

| Model design | | | Total effect | | Mediation summary | | | | Peripheral vascular reactivity (PVR) | | | | Cerebrovascular reactivity (CVR) | | | | PVR → CVR | |
| | | | | | Indirect effect | | Direct effect | | A1 path (x → PVR) | | B1 path (PVR → y) | | A2 path (x → CVR) | | B2 path (CVR → y) | | M1 – M2 path | |
| M1 | M2 | n | Est | P | Est | P | Est | P | Est | P | Est | P | Est | P | Est | P | Est | P |
|---|---|---|---|---|---|---|---|---|---|---|---|---|---|---|---|---|---|---|
| Endothelium-dependent reactivity (ACh) | CBF arrival time normocapnia | 39 | −0.58 | <0.001 | 0.00 [0.00; 0.00] | 0.98 | −0.50 [−0.72; −0.27] | <0.001 | −0.31 [−0.59; −0.04] | 0.01 | 0.21 [−0.05; 0.47] | 0.11 | −0.05 [−0.38; 0.28] | 0.12 | 0.00 [−0.25; 0.25] | 0.98 | −0.03 [−0.36; 0.3] | 0.92 |
| | CBF arrival time hypercapnia | 38 | −0.60 | <0.001 | 0.00 [−0.02; 0.01] | 0.93 | −0.50 [−0.74; −0.26] | <0.001 | −0.35 [−0.62; −0.08] | 0.01 | 0.26 [−0.01; 0.52] | 0.06 | 0.32 [0.02; 0.62] | 0.04 | 0.13 [−0.13; 0.40] | 0.32 | 0.01 [−0.31; 0.34] | 0.93 |
| | CBF arrival time hypocapnia | 38 | −0.56 | <0.001 | 0.00 [−0.01; 0.01] | 0.79 | −0.48 [−0.71; −0.25] | <0.001 | −0.33 [−0.62; −0.05] | 0.02 | 0.24 [−0.02; 0.51] | 0.07 | 0.07 [−0.26; 0.41] | 0.68 | 0.06 [−0.19; 0.32] | 0.64 | 0.06 [−0.28; 0.39] | 0.73 |
| | CBF perfusion normocapnia | 39 | −0.58 | <0.001 | 0.01 [−0.01; 0.01] | 0.69 | −0.45 [−0.67; −0.23] | <0.001 | −0.31 [−0.59; −0.04] | 0.01 | 0.23 [−0.01; 0.48] | 0.06 | −0.16 [−0.48; 0.16] | 0.28 | 0.30 [0.06; 0.53] | 0.02 | −0.08 [−0.4; 0.25] | 0.68 |
| | CBF perfusion hypercapnia | 38 | −0.60 | <0.001 | 0.00 [−0.03; 0.04] | 0.75 | −0.43 [−0.68; −0.19] | <0.001 | −0.35 [−0.62; −0.08] | 0.01 | 0.27 [0.00; 0.53] | 0.05 | −0.17 [−0.5; 0.16] | 0.32 | 0.15 [−0.10; 0.40] | 0.25 | −0.06 [−0.39; 0.28] | 0.74 |
| | CBF perfusion hypocapnia | 38 | −0.56 | <0.001 | 0.00 [−0.01; 0.02] | 0.81 | −0.41 [−0.64; −0.17] | 0.001 | −0.33 [−0.6; −0.05] | 0.02 | 0.26 [0.01; 0.51] | 0.04 | −0.24 [−0.56; 0.07] | 0.13 | 0.28 [0.04; 0.52] | 0.03 | −0.04 [−0.37; 0.29] | 0.81 |
| | CVRhyper whole brain | 38 | −0.60 | <0.001 | 0.00 [−0.03; 0.03] | 0.97 | −0.46 [−0.69; −0.23] | <0.001 | −0.35 [−0.62; −0.08] | 0.01 | 0.26 [0.00; 0.52] | 0.05 | −0.01 [−0.35; 0.33] | 0.96 | −0.17 [−0.42; 0.08] | 0.18 | −0.01 [−0.35; 0.33] | 0.97 |
| | CVRhyper V1 | 39 | −0.58 | <0.001 | −0.01 [−0.05; 0.03] | 0.66 | −0.44 [−0.66; −0.22] | <0.001 | −0.35 [−0.61; −0.08] | 0.01 | 0.21 [−0.03; 0.45] | 0.09 | 0.09 [−0.24; 0.42] | 0.60 | −0.35 [−0.57; −0.13] | 0.002 | −0.08 [−0.41; 0.25] | 0.65 |
| | CVRhypo V1 | 38 | −0.56 | <0.001 | 0.00 [−0.01; 0.02] | 0.59 | −0.46 [−0.7; −0.23] | <0.001 | −0.33 [−0.6; −0.05] | 0.02 | 0.26 [0.00; 0.53] | 0.05 | 0.14 [−0.19; 0.46] | 0.42 | −0.08 [−0.34; 0.17] | 0.53 | 0.19 [−0.14; 0.51] | 0.25 |
| Endothelium-Independent reactivity (SNP) | CBF arrival time normocapnia | 39 | −0.58 | <0.001 | 0.00 [−0.01; 0.01] | 0.98 | −0.50 [−0.73; −0.28] | <0.001 | −0.31 [−0.59; −0.03] | 0.02 | 0.19 [−0.08; 0.45] | 0.14 | −0.06 [−0.39; 0.27] | 0.11 | 0.01 [−0.25; 0.26] | 0.98 | −0.07 [−0.4; 0.26] | 0.91 |
| | CBF arrival time hypercapnia | 38 | −0.60 | <0.001 | 0.00 [−0.01; 0.01] | 0.75 | −0.53 [−0.77; −0.3] | <0.001 | −0.32 [−0.6; −0.04] | 0.02 | 0.19 [−0.07; 0.46] | 0.16 | 0.30 [0.00; 0.60] | 0.05 | 0.15 [−0.12; 0.42] | 0.29 | −0.06 [−0.37; 0.26] | 0.73 |
| | CBF arrival time hypocapnia | 38 | −0.56 | <0.001 | 0.00 [−0.01; 0.02] | 0.73 | −0.51 [−0.74; −0.28] | <0.001 | −0.30 [−0.58; −0.02] | 0.04 | 0.16 [−0.11; 0.44] | 0.23 | 0.10 [−0.23; 0.42] | 0.56 | 0.05 [−0.21; 0.31] | 0.70 | 0.15 [−0.18; 0.47] | 0.37 |
| | CBF perfusion normocapnia | 39 | −0.58 | <0.001 | 0.00 [−0.03; 0.03] | 0.72 | −0.47 [−0.69; −0.24] | <0.001 | −0.31 [−0.59; −0.03] | 0.02 | 0.19 [−0.06; 0.44] | 0.09 | −0.14 [−0.46; 0.19] | 0.28 | 0.28 [0.05; 0.52] | 0.02 | −0.01 [−0.34; 0.32] | 0.71 |
| | CBF perfusion hypercapnia | 38 | −0.60 | <0.001 | 0.00 [−0.01; 0.01] | 0.99 | −0.47 [−0.71; −0.23] | <0.001 | −0.32 [−0.59; −0.03] | 0.02 | 0.18 [−0.08; 0.45] | 0.18 | −0.15 [−0.47; 0.18] | 0.38 | 0.13 [−0.12; 0.39] | 0.31 | 0.00 [−0.33; 0.33] | 0.99 |
| | CBF perfusion hypocapnia | 38 | −0.56 | <0.001 | 0.01 [−0.02; 0.05] | 0.39 | −0.42 [−0.66; −0.18] | 0.001 | −0.30 [−0.58; −0.02] | 0.04 | 0.22 [−0.03; 0.48] | 0.09 | −0.28 [−0.58; 0.02] | 0.07 | 0.30 [0.06; 0.55] | 0.02 | −0.17 [−0.48; 0.15] | 0.30 |
| | CVRhyper whole brain | 38 | −0.60 | <0.001 | 0.00 [−0.02; 0.03] | 0.55 | −0.48 [−0.71; −0.26] | <0.001 | −0.32 [−0.6; −0.04] | 0.02 | 0.21 [−0.06; 0.47] | 0.12 | 0.03 [−0.3; 0.37] | 0.85 | −0.19 [−0.45; 0.06] | 0.13 | 0.12 [−0.21; 0.45] | 0.49 |
| | CVRhyper V1 | 39 | −0.58 | <.001 | 0.00 [−0.04; 0.03] | 0.79 | −0.46 [−0.68; −0.24] | <0.001 | −0.31 [−0.59; −0.04] | 0.03 | 0.15 [−0.09; 0.39] | 0.23 | 0.1 [−0.22; 0.43] | 0.54 | −0.36 [−0.58; −0.13] | 0.002 | −0.04 [−0.37; 0.28] | 0.79 |
| | CVRhypo V1 | 38 | −0.56 | <.001 | 0.00 [0.00; 0.00] | 0.89 | −0.50 [−0.73; −0.27] | <0.001 | −0.30 [−0.58; −0.02] | 0.04 | 0.17 [−0.10; 0.44] | 0.20 | 0.08 [−0.25; 0.41] | 0.63 | −0.04 [−0.3; 0.22] | 0.75 | 0.03 [−0.3; 0.36] | 0.87 |

No combination of M1 and M2 variables significantly explained the relationship between platelet sensitivity (x = sensitivity to ADP) and the duration of the neurovascular response (y = HRF FWHM). CBF, cerebral blood flow; CVRhyper, hypercapnic cerebrovascular reactivity; CVRhypo, hypocapnic cerebrovascular reactivity; Est, estimated effect; SNP, sodium nitroprusside (nitric oxide donor); V1, primary visual cortex.

cascade of detrimental events that contribute to the development and progression of dementia.

Although the functional implications of these novel findings are not yet fully understood, the observed effects in our study are consistent with previous literature demonstrating altered cerebral neurovascular coupling in clinical states. For example, subjective cognitive decline has been associated with shorter FWHM and longer TTP compared to age-matched controls (Lu et al., 2022). Older age has been associated with a longer TTP (West et al., 2019), whereas diabetes has also been associated with both a longer TTP and a smaller AUC (Guimarães et al., 2024; Van Duarte et al., 2015). These previous studies also reported smaller HRF peak heights compared to controls, but there were no associations between platelet reactivity and HRF peak height in the present study. We also found no effect of age on any HRF parameters, which contrasts with previous work that specifically investigated age-effects in healthy adults (West et al., 2019). However, this discrepancy is reconciled by the fact that West and colleagues compared a group of younger adults (18–30 years) to a group of older adults (54–74 years), whereas the present study investigated a continuous effect of age within older adults only (50–80 years).

It is interesting that platelet reactivity to specific agonists was related to specific components of the haemodynamic response. Platelet aggregation in response to ADP was specifically related to narrower HRF FWHM, whereas platelet aggregation in response to TRAP-6 was specifically associated with smaller HRF AUC, and platelet aggregation in response to CRP was specifically related to slower TTP. ADP is released by activated platelets at the site of injury and potentiates platelet aggregation by binding to purinergic receptors on platelets, leading to a conformational change of the integrin $\alpha IIb\beta 3$ on the platelet surface, allowing fibrinogen binding and platelet aggregation (Gachet, 2006). This autocrine positive feedback loop amplifies platelet aggregation and thrombus formation. By contrast, TRAP-6 is a peptide that mimics the effects of thrombin and promotes platelet activation and aggregation via activation of protease-activated receptors on the platelet surface (Jensen et al., 2013). CRP is a cross-linked peptide that mimics the action of collagen, which is exposed upon vascular injury. CRP (and collagen) initiates platelet adhesion and activation by binding to the glycoprotein VI (GPVI) receptor on platelets (Farndale, 2006). These differing roles in the haemostasis cascade provide scope for future mechanistic studies to investigate, although it is worth noting that the only relationship remaining after correction for multiple comparisons was platelet sensitivity to ADP; the only agonist from the platelet assay that is released by platelets *in vivo*.

In this study, we focused on platelet reactivity assays to indicate platelet function in different physiological contexts. Platelet reactivity provides a more direct measure of platelet responses to specific physiological stimuli, compared to platelet activation markers which primarily reflect static activation status or systemic inflammation. Unlike activation markers, which may be sensitive to acute inflammatory events (e.g. infections or injuries), platelet reactivity assays are less influenced by external stimuli.

Although our study focused on platelet reactivity, it is important to consider how activated platelets could impact neurovascular function through various mechanisms. Activated platelets release a range of factors that influence vascular tone, blood–brain barrier (BBB) integrity and neuroinflammation, which can collectively affect cerebral circulation and neural function. Platelets may influence neurovascular unit by altering vascular integrity. Platelets release vascular endothelial growth factor (Salgado et al., 2001), and transforming growth factor-beta (TGF-$\beta$) $\beta$ (Assoian & Sporn, 1986), both of which can directly increase BBB permeability, particularly under hypoxic conditions (Zhang et al., 2000). In addition, TGF-$\beta$ has been shown to alter the metabolic behaviour of pericytes, shifting their energy production from oxidative phosphorylation to glycolysis (Schumacher et al., 2023). Pericytes play a key role in maintaining the structural and functional integrity of the BBB, as well as regulating capillary flow, angiogenesis, and immune responses (Brown et al., 2019; Sá-Pereira et al., 2012; Sweeney et al., 2016). Furthermore, platelet-derived growth factor BB (PDGF-BB) signalling promotes pericyte recruitment, proliferation and migration, contributing to vascular tone regulation and neurovascular function (Gaceb et al., 2018).

Platelet activation is associated with the release of inflammatory mediators that can lead to secondary BBB disruption, immune cell recruitment, and altered CBF. The release of P-selectin and chemokines, including $\beta$-thromboglobulin and platelet factor 4, can promote platelet-leukocyte interactions and contribute to endothelial dysfunction (Bakogiannis et al., 2019; Kutlar & Embury, 2014; Totani & Evangelista, 2010), potentially impairing vascular integrity and blood flow. Interestingly, recent research has shown that administering platelet factor 4 from young mice improved cognition and reduced neuroinflammation in the hippocampus of aged mice (Schroer et al., 2023).

Platelet-derived mediators can activate sensory neurons and contribute to neurogenic inflammation. Platelets express receptors for tachykinins, such as substance P and neurokinin 1. These neuropeptides act on neurovascular structures to modulate inflammation and cerebrovascular responses. Research by our group has demonstrated their additional role in platelet thrombus formation and regulation (Graham et al., 2004; Jones et al., 2008). This provides another biological overlap between platelets and

the neurovascular unit that extends beyond a systemic contribution to atherosclerosis.

Finally, platelets may affect cerebral neurovascular function through neurotransmitter-mediated vaso-regulation. Serotonin release by activated platelets (Mercado & Kilic, 2010) may link platelet activity to neurovascular function by modulating neuro-transmission and vascular tone. Serotonin acts on receptors present on neurons, glial cells and vascular smooth muscle cells (Hoyer et al., 1994) and can induce vasoconstriction, thereby altering blood flow and neural activity (Vanhoutte, 1987). This connection between platelets and vascular tone could influence the regulation of blood flow within the neurovascular unit, impacting both CBF and neuronal function.

These potential mechanisms, although speculative in the context of our study, outline the large degree of biological overlap between platelet regulatory mechanisms and the neurovascular unit. Future research should incorporate both functional reactivity measures and the mechanistic roles of platelet-released factors to further elucidate how platelet reactivity impacts the neurovascular unit and cognitive health.

Not only was the relationship between platelet and cerebral neurovascular function not explained by systemic vascular function, but also there was no evidence for consistent 'systemic vascular function' (i.e. agreement between peripheral and cerebral vascular reactivity). This is in line with what others have observed (Palazzo et al., 2013), although the previous literature compared large blood vessels, whereas the present study specifically investigated microvascular perfusion in both the arm and the brain. This could be a result of physiological differences between the peripheral vasculature and cerebrovasculature, or could simply be due to different vasoactive experimental manipulations (NO in the periphery *vs.* $CO_2$ in the cerebrum).

Cerebrovascular function (assessed through cerebrovascular reactivity) and cerebral neurovascular function (assessed through HRF parameters) have shared physiological bases. However, to our knowledge, this is the first study to demonstrate a relationship between these measures in humans. HRF FWHM was positively associated with multiple (non-neuronal) cerebrovascular measures; CBF in normocapnia, CBF in hypocapnia and cerebrovascular reactivity to hypercapnia (CVRhyper). It is particularly interesting that the duration of the haemodynamic response (HRF FWHM) was associated with the magnitude of the blood flow response to hyper-capnia. Although the initiation of the BOLD response is purported to relate to synaptically-driven feedforward signalling, the sustained BOLD response is purported to relate to feedback signalling driven more by $CO_2$ and waste products (Iadecola, 2017). This finding therefore provides important empirical evidence to support this mechanistic picture, using integrated functional measures in humans.

This study is limited by its reliance on the BOLD response to infer cerebral neurovascular function. It is possible the reduced haemodynamic responses observed may reflect diminished underlying neuronal activity with intact cerebral neurovascular coupling because the BOLD signal is unable to differentiate between the neural and vascular components of cerebral neuro-vascular function. However, it is unclear through what mechanism higher platelet reactivity could affect neuro-nal activity isolated from cerebral neurovascular function. By contrast, as outlined above, there are plausible physio-logical mechanisms that could explain an association between platelet reactivity and cerebral neurovascular function. Another limitation of this study is the use of a cross-sectional design. A longitudinal design would have provided evidence for directionality, as well as insights on the functional implications of the observed relationships, allowing us to map the trajectory of potential neurodegeneration. Our findings indicate that such investigation would help not only further under-stand but also capitalise on the phenomenon observed herein. This present study also limited by the focus on the visual cortex (V1) and did not include measures of cognitive function, or performance-based tasks, but the dataset produced by our broader project will allow us and others to address some of these questions.

In conclusion, we show that platelet reactivity was associated with cerebral neurovascular function in a healthy older sample. This novel association appears to bypass systemic vascular function and to be mechanistically specific; distinct platelet signalling mechanisms were associated with different physio-logical components of the haemodynamic response to neural activity. Altered cerebral neurovascular function (haemodynamic responses to neuronal activation) marks the initial physiological stage in a cascade of events contributing to neurodegeneration. Therefore, under-standing the precise mechanisms responsible for this novel finding may provide a substantial step forward in preventing this global health concern.

# Appendix

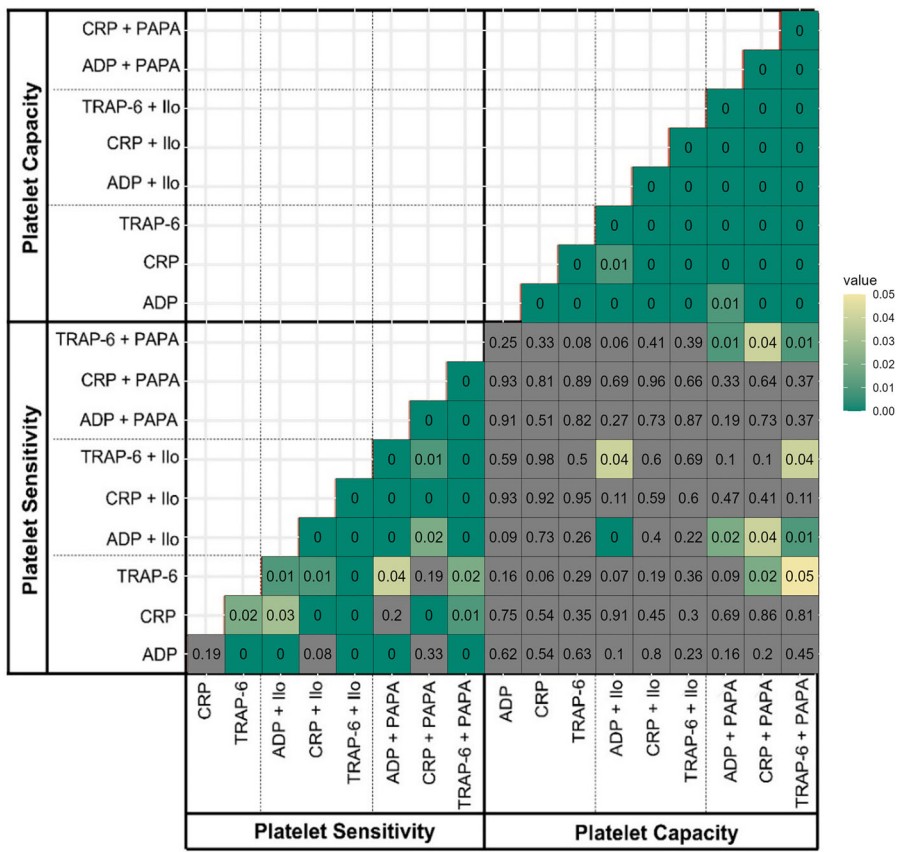

**Figure A1. Matrix of *P* values within platelet reactivity metrics**
Corresponding to Spearman's rho values presented in Fig. 1*B*. Correlations between platelet reactivity measures confirmed sensitivity and capacity are distinct concepts. [Colour figure can be viewed at wileyonlinelibrary.com]

### *A* Excluded HRFs

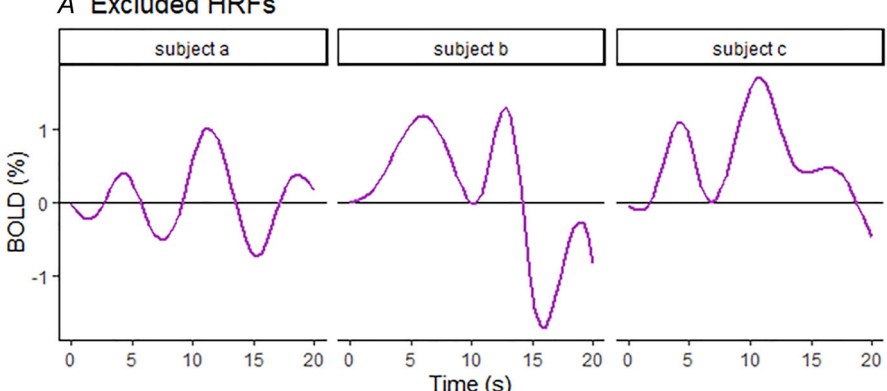

### *B* Constraining HRFs by local minima

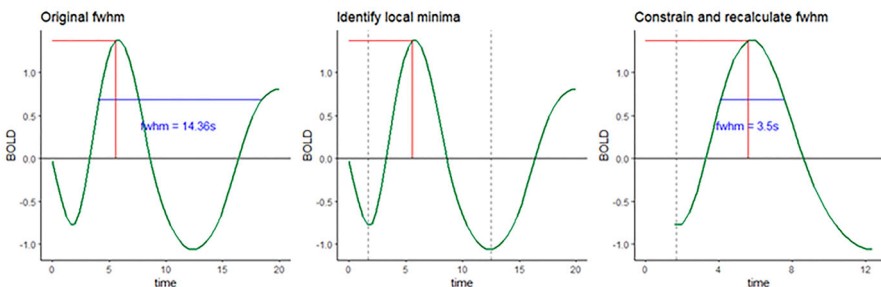

### *C* Calculating area under the curve (AUC)

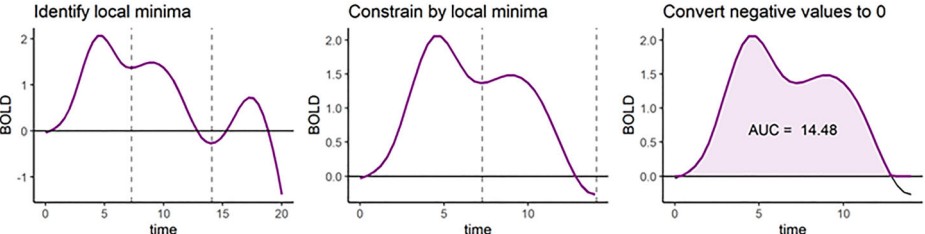

**Figure A2. Schematic representation of qualitative assessment procedure for haemodynamic response functions (HRFs)**

*A*, HRF time series that were not physiologically plausible, or where the HRF could not be confidently identified, were excluded on the basis any extracted HRF parameters would reflect only noise. *B*, HRF parameters were calculated systematically but constrained by local minima corresponding to qualitative assessment of a single HRF impulse. This was necessary because the duration covered by HRF time series was set at 20 s across all cases, but clearly covered multiple HRF impulses in some cases. [Colour figure can be viewed at wileyonlinelibrary.com]

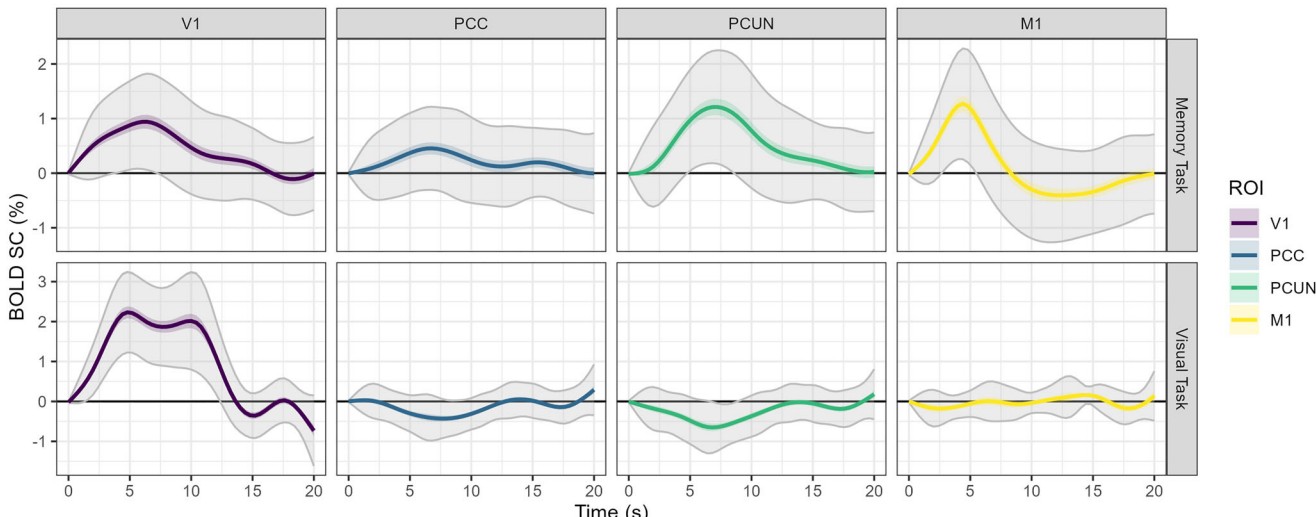

**Figure A3. Group-averaged task-evoked haemodynamic response functions (HRFs) from multiple tasks and brain regions**

Only the V1 HRFs from the visual task were included in analyses incorporating other variables (platelet reactivity and vascular reactivity). All HRFs included in the main study are presented here as a manipulation check for the HRF analysis. Solid lines represent the group mean, coloured ribbons indicate the SE, and grey ribbons indicate the SD. Data from all participants are included in these figures, including haemodynamic traces from individuals with non-physiologically plausible HRFs, which were excluded from subsequent HRF parameter analyses. Group-level HRFs were observed only in *a priori* task-relevant regions of interest (ROIs). M1, motor cortex (left, digit 1); PCC, posterior cingulate cortex; PCUN, precuneus; V1, primary visual cortex. [Colour figure can be viewed at wileyonlinelibrary.com]

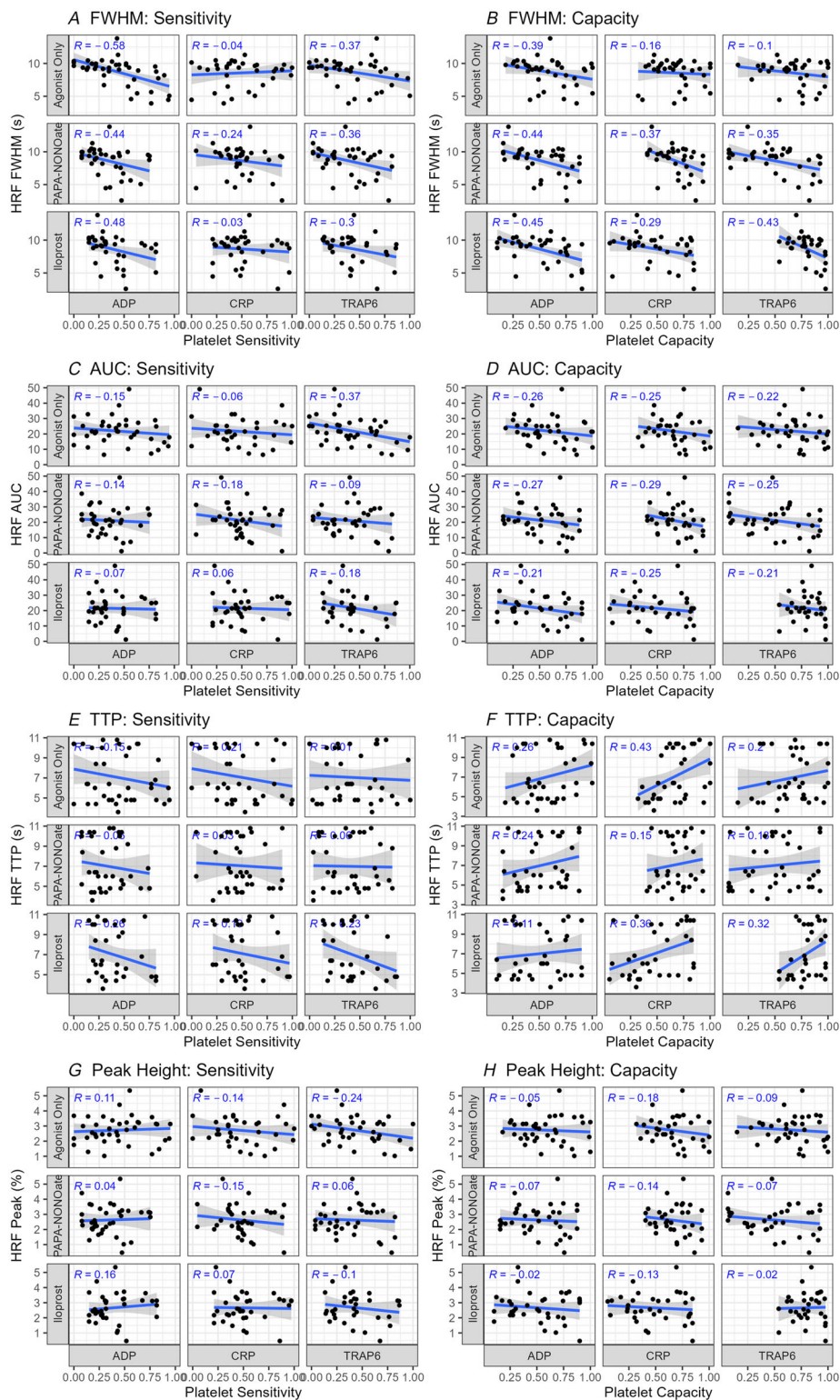

**Figure A4. Scatter plots**

All correlations presented as heatmaps in Fig. 4 are presented as full scatter plots displaying the individual data points here. A, relationships between platelet sensitivity (EC50) and HRF full-width half-maximum (FWHM). B, relationships between platelet capacity (Emax) and HRF FWHM. C, relationships between platelet sensitivity and HRF area under the curve (AUC). D, relationships between platelet capacity and HRF AUC. E, relationships between platelet sensitivity and HRF time to peak (TTP). F, relationships between platelet capacity and HRF TTP. G, relationships between platelet sensitivity and HRF peak height. H, relationships between platelet capacity and HRF peak height [Colour figure can be viewed at wileyonlinelibrary.com]

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

## Additional information

### Data availability statement

Pre-processed data is provided with this manuscript. Raw (defaced) MRI data files are available on OpenNeuro (https://openneuro.org).

### Competing interests

The authors declare that they have no competing interests.

### Author contributions

G.M.K.R., A.C., J.M.G., J.L.D. and J.A.L. were responsible for the conception and design of the work. G.M.K.R., A.S., B.W., K.M.C., S.R., E.J. and E.B. were responsible for data acquisition. G.M.K.R., J.L.D., A.S., E.J. were responsible for data analysis. G.M.K.R., J.L.D., J.M.G., A.C. were responsible for writing the original draft. G.M.K.R., J.L.D., A.S., B.W., K.M.C., S.R., E.J., E.B., J.A.L., J.M.G. and A.C. were responsible for the reviewing and editing.

### Funding

The study was funded by the University of Reading. JLD and JMG thank the British Heart Foundation for support (RG/20/7/34 866).

### Acknowledgements

We thank Shan Shen for her technical assistance. This manuscript was first published as a preprint: Rossetti GMK, Dunster JL. Sohail A, Williams B, Cox KM, Jewett E, Benford E, Lovegrove JA, Gibbins JM, Christakou A. (2024). Evidence for direct control of neurovascular function by circulating platelets in healthy older adults. bioRxiv. https://www.biorxiv.org/content/10.1101/2024.05.31.596788v1.

### Keywords

ageing, cerebral blood flow, neuroscience, magnetic resonance imaging, platelets, vascular physiology

## Supporting information

Additional supporting information can be found online in the Supporting Information section at the end of the HTML view of the article. Supporting information files available:

**Peer Review History**
**Supplementary Table**
**Table S1**
**Data S1**

