## [Peer Review History · The Journal of Physiology]

Evidence for control of cerebral neurovascular function by circulating platelets in healthy older adults

Gabriella Maria Knasel Rossetti, Joanne L Dunster, Aamir Sohail, Brendan Williams, Kiera M Cox, Suzannah Rawlings, Elysia Jewett, Eleanor Benford, Julie A Lovegrove, Jonathan M Gibbins, and Anastasia Christakou
DOI: 10.1113/JP288405

Corresponding author(s): Gabriella Rossetti (g.rossetti@mmu.ac.uk)

The following individual(s) involved in review of this submission have agreed to reveal their identity: Luke A Henderson (Referee #1)

Review Timeline:

Submission Date:	07-Jan-2025
Editorial Decision:	25-Feb-2025
Revision Received:	03-Apr-2025
Accepted:	10-Apr-2025

Senior Editor: Harold Schultz

Reviewing Editor: Vaughan Macefield

Transaction Report:

Dear Dr Rossetti,

Re: JP-RP-2025-288405 "Evidence for direct control of neurovascular function by circulating platelets in healthy older adults" by Gabriella Maria Knasel Rossetti, Joanne L Dunster, Aamir Sohail, Brendan Williams, Kiera M Cox, Elysia Jewett, Eleanor Benford, Julie A Lovegrove, Jonathan M Gibbins, and Anastasia Christakou

Thank you for submitting your manuscript to The Journal of Physiology. It has been assessed by a Reviewing Editor and by 2 expert referees and we are pleased to tell you that it is acceptable for publication following satisfactory revision.

REVISION CHECKLIST:

We look forward to receiving your revised submission.

Yours sincerely,

Harold Schultz
Senior Editor
The Journal of Physiology

EDITOR COMMENTS

Reviewing Editor:

Comments to the Author:

Thank you for submitting your manuscript to The Journal of Physiology. I have now received comments from two independent expert reviewers. As you will see from their favourable comments, they believe your study uncovers some interesting and important physiology. In comments to me, the reviewers believe that your results "will be influential in the field of neurovascular coupling and have an influence beyond that field" and that "exploring the connection between platelet activation and early neurovascular dysfunction in the brain is a needed and potentially important area of research."

However, there are some issues you will need to address before we can consider the manuscript further. I invite you to revise the manuscript accordingly and submit point-by-point responses to the reviewers' comments. Moreover, you will need to reformat the article to meet the publication requirements of the Journal, so please consult the Submission Guidelines on our website. For example, references should be cited in the text and listed alphabetically in the references list (not numbered according to citation) and the Methods section needs to follow the Introduction. Furthermore, the Journal does not publish supplementary material, so please incorporate all of your essential supplementary material in the body of the manuscript.

I look forward to receiving your revised manuscript in due course.

Senior Editor:

Comments to the Author:

Thank you for submission of your research article to the Journal of Physiology for consideration. The article has been reviewed by experts in the field and found to be potentially acceptable for publication pending adequate revision to address all of the concerns raised. Please address all comments from the external referees and reviewing editor as well as addressing the list of requirements or publication in the journal including the following:

1. The term "neurovascular function" used throughout the manuscript is too ambiguous for our journal readership, whose focus encompasses peripheral as well as cerebral neurovascular function. Please incorporate appropriate references to cerebral neurovascular function in the title, key points, abstract, and throughout the text to avoid confusion to the readers. This pertains also to other references to cerebral vascular function such as cerebral blood flow, etc.
2. The citations and bibliography do not follow the journal format.
3. Methods must follow the Introduction.
3. The Methods must begin with the Ethical Approval section.
4. Ensure that all Figures and Tables are cited.
4. The journal does not allow supplements of information relevant to understanding the study. This requirement is for the

benefit of our readers to allow this information to be contained in the article PDF. Please incorporate the supplement methods into the Methods section. The supplementary figures can be incorporated into the Results section as an Appendix after the Discussion, cited as "see Appendix, Fig xx).

5. Provide other journal requirements such as Abstract Figure and legend and first author profile.

REFeree COMMENTS

Referee #1:

This manuscript details the relationship between neurovascular coupling and platelet function in humans. Overall, the manuscript is well written, the design appropriate, the analysis appropriate and the findings are extremely interesting and could have significant consequences not only in the field of aging and dementia but in brain imaging more generally. The finding that higher platelet reactivity is associated with shorter and smaller hemodynamic response functions is important and novel and makes one rethink the relationships that govern bold signal intensity changes. The introduction and Discussion are comprehensive and well-written. Dividing participants into 3 groups based on platelet phenotype reduces the overall statistical power of the study although the relationships that are not significant are not even close to significance and so additional participants is unlikely to alter the overall results. I have very little to offer with respect to feedback as I suggest the study is appropriate for publication in its current form.

Minor points:

- 1) the identification of hemodynamic BOLD changes relies on visual inspection and a determination that the changes are not physiological. I do find this a little peculiar and would have thought that the response was fairly robust. Maybe this should be discussed.
- 2) It is unclear to me exactly how the hemodynamic response was calculated. Was it the average over all 6.8s trials. I understand the FIR function and was wondering why not simply extract V1 signal changes for each visual stimulus trial and average it for each individual?

Referee #2:

The manuscript by G Rossetti titled "Evidence for direct control of neurovascular function by circulating platelets in healthy older adults" investigates the link between platelet reactivity and neurovascular function in healthy middle-aged and older adults. The authors employed a multiparameter phenotyping approach using plate-based aggregometry to assess platelet function from human blood samples. They measured responses to different platelet agonists, including ADP, Thrombin Receptor Activator Peptide 6 (TRAP-6), and Collagen-Related Peptide (CRP). They also did this in the presence of inhibitors like a nitric oxide donor and a prostacyclin analog. In parallel, participants underwent functional MRI during a visual stimulation task (flashing checkerboard) to evoke blood flow responses in the primary visual cortex. From these data, the authors extracted the hemodynamic response function parameters, including full-width half maximum, area under the curve, time to peak, and peak height. The results showed that higher platelet reactivity, especially sensitivity to ADP, correlated with a significantly narrower FWHM and reduced AUC neurovascular response. The authors ruled out systemic vascular reactivity, assessed through both peripheral and cerebrovascular measures, as the cause, suggesting a direct platelet effect on the neurovascular unit. Clustering analyses further delineated distinct platelet phenotype groups, reinforcing this association. The authors propose that platelets influence neurovascular coupling and may contribute to early neurodegenerative changes.

The strengths of the paper include an interesting integration of platelet functional assays with fMRI measures of neurovascular function; a comprehensive statistical analyses including principal component analysis, clustering, and mediation models; and a control for demographic confounders. The data presentation is excellent and the figures overall are of high quality. The writing is clear.

The authors also do a good job of listing and describing the limitations of this study in the discussion, such as a focus on only visual cortex responses, no behavioural or cognitive relationships explored to platelet activation, and limits of BOLD signal itself in not being able to distinguish between changes in neural activity vs changes in vascular function.

Regarding weaknesses, I have no major problems with this work, but a couple points:

There was a limited exploration of platelet activation markers and potential molecular mechanisms. Can the authors justify their choices versus some others, such as looking at 1) P-selectin, 2) Integrin $\alpha\text{IIb}\beta\text{3}$, or 3) the release of Beta-Thromboglobulin or Platelet Factor 4? This could just be a discussion piece.

I would like the authors to refine some of the wording regarding the relationship between platelet factors and NVU function. As the data are primarily correlational, wording like "direct link" or "direct association" should drop the "direct". What is the difference between a direct association and an association? I believe "direct" is unnecessary and potentially misleading/inaccurate. When talking about links or connections, "potential link" should be used instead.

END OF COMMENTS

01/04/2025

Dear Journal of Physiology

RE: JP-RP-2025-288405 (*Evidence for control of cerebral neurovascular function by circulating platelets in healthy older adults*)

On behalf of my co-authors, I would like to thank the editors and the reviewers for their time reviewing the manuscript, their positive comments, and their helpful suggestions.

We have incorporated all their suggestions and responded to each of their comments individually below. I have included the original comment text in black and our responses below in blue.

Kind Regards
Gabriella Rossetti
g.rossetti@mmu.ac.uk

EDITOR COMMENTS

Reviewing Editor:

Comments to the Author:

Thank you for submitting your manuscript to The Journal of Physiology. I have now received comments from two independent expert reviewers. As you will see from their favourable comments, they believe your study uncovers some interesting and important physiology. In comments to me, the reviewers believe that your results "will be influential in the field of neurovascular coupling and have an influence beyond that field" and that "exploring the connection between platelet activation and early neurovascular dysfunction in the brain is a needed and potentially important area of research."

However, there are some issues you will need to address before we can consider the manuscript further. I invite you to revise the manuscript accordingly and submit point-by-point responses to the reviewers' comments. Moreover, you will need to reformat the article to meet the publication requirements of the Journal, so please consult the Submission Guidelines on our website. For example, references should be cited in the text and listed alphabetically in the references list (not numbered according to citation) and the Methods section needs to follow the Introduction. Furthermore, the Journal does not publish supplementary material, so please incorporate all of your essential supplementary material in the body of the manuscript.

I look forward to receiving your revised manuscript in due course.

Senior Editor:

Comments to the Author:

Thank you for submission of your research article to the Journal of Physiology for consideration. The article has been reviewed by experts in the field and found to be potentially acceptable for publication pending adequate revision to address all of the concerns raised. Please address all comments from the external referees and reviewing editor as well as addressing the list of requirements or publication in the journal including the following:

1. The term "neurovascular function" used throughout the manuscript is too ambiguous for our journal readership, whose focus encompasses peripheral as well as cerebral neurovascular function. Please incorporate appropriate references to cerebral neurovascular function in the title, key points, abstract, and throughout the text to avoid confusion to the readers. This pertains also to other references to cerebral vascular function such as cerebral blood flow, etc.

Thank you for highlighting this point, we have changed references to neurovascular function and neurovascular coupling to be clear that they are specific to cerebral throughout.

2. The citations and bibliography do not follow the journal format.

This has been amended, both now follow the journal format.

3. Methods must follow the Introduction.

This has been moved and checked for flow. Any necessary changes to ensure the narrative flows logically have been made.

3. The Methods must begin with the Ethical Approval section.

This has been moved.

4. Ensure that all Figures and Tables are cited.

All figures and tables are now cited, including figures previously included only as supplementary (and formatting updated to ensure they are linked).

5. The journal does not allow supplements of information relevant to understanding the study. This requirement is for the benefit of our readers to allow this information to be contained in the article PDF. Please incorporate the supplement methods into the Methods section. The supplementary figures can be incorporated into the Results section as an Appendix after the Discussion, cited as "see Appendix, Fig xx).

Thank you for this suggestion, and I appreciate this is beneficial for the readers. We have followed your suggestion.

6. Provide other journal requirements such as Abstract Figure and legend and first author profile.

The abstract figure and first author profile have been prepared and are included in this submission

Referee #1:

This manuscript details the relationship between neurovascular coupling and platelet function in humans. Overall, the manuscript is well written, the design appropriate, the analysis appropriate and the findings are extremely interesting and could have significant consequences not only in the field of aging and dementia but in brain imaging more generally. The finding that higher platelet reactivity is associated with shorter and smaller hemodynamic response functions is important and novel and makes one rethink the relationships that govern bold signal intensity changes. The introduction and Discussion are comprehensive and well-written. Dividing participants into 3 groups based on platelet phenotype reduces the overall statistical power of the study although the relationships that are not significant are not even close to significance and so additional participants is unlikely to alter the overall results. I have very little to offer with respect to feedback as I suggest the study is appropriate for publication in its current form.

Minor points:

1) the identification of hemodynamic BOLD changes relies on visual inspection and a determination that the changes are not physiological. I do find this a little peculiar and would have thought that the response was fairly robust. Maybe this should be discussed.

You are correct that the response in the visual cortex is typically robust, although more variable in shape than often suggested in the literature. You are of course right to pick up that this step seems odd within the context of this manuscript. The reason for this is that our HRF determination and analysis technique was designed for the broader study, which also assessed brain regions outside primary sensory cortex during cognitive tasks, for example, the posterior cingulate cortex [PCC] and precuneus during memory recall.

This has been addressed concisely in the methods for the benefit of the reader, and is explained here in more detail for your convenient reference.

In the methods:

This approach was necessary for the broader study, which included cognitive brain regions beyond the visual cortex, where the robustness of the HRF varied.

For the reviewer:

Outside the primary visual cortex, we found in a handful of individuals some responses that do not reflect the physiologically canonical HRF. These cases were excluded on the basis that extracting the peak height, time to peak, FWHM, and AUC would not be meaningful parameters to include in subsequent analysis. Two examples of this data (from the PCC) are included below for your reference.

In some cases we observed negative BOLD responses, despite adequate performance on the task, which will be investigated in a subsequent paper. Of note, we read with interest this preprint article by Epp and colleagues (Epp et al., preprint doi:10.1101/2023.12.08.570806).

However, this was not the case for the V1 where, as you state, the response is much more robust. The excluded HRFs for the V1 are included in the Appendix (Figure 7A) and below (for your reference). These consisted of 3 individuals where for each 20 second window, two distinct peaks could be identified, meaning a single individual HRF could not be identified from the 20s window.

We have chosen to separate the findings from the visual task and the memory task into separate papers in order to make each digestible for the reader. However, in the interest of rigour, we do account for these analyses in our correction for multiple comparisons.

2) It is unclear to me exactly how the hemodynamic response was calculated. Was it the average over all 6.8s trials. I understand the FIR function and was wondering why not simply extract V1 signal changes for each visual stimulus trial and average it for each individual? As with above, the approach was designed to be consistent with the analysis of the memory task data. The memory recall task required a more complicated GLM that necessitated the use of the FIR approach, and we applied this same approach to the visual task data, despite the fact that V1 signal changes from each visual stimulus would have been sufficient for this task.

Referee #2:

The manuscript by G Rossetti titled "Evidence for direct control of neurovascular function by circulating platelets in healthy older adults" investigates the link between platelet reactivity and neurovascular function in healthy middle-aged and older adults. The authors employed a multiparameter phenotyping approach using plate-based aggregometry to assess platelet function from human blood samples. They measured responses to different platelet agonists, including ADP, Thrombin Receptor Activator Peptide 6 (TRAP-6), and Collagen-Related Peptide (CRP). They also did this in the presence of inhibitors like a nitric oxide donor and a prostacyclin analog. In parallel, participants underwent functional MRI during a visual stimulation task (flashing checkerboard) to evoke blood flow responses in the primary visual cortex. From these data, the authors extracted the hemodynamic response function parameters, including full-width half maximum, area under the curve, time to peak, and peak height. The results showed that higher platelet reactivity, especially sensitivity to ADP, correlated with a significantly narrower FWHM and reduced AUC neurovascular response. The authors ruled out systemic vascular reactivity, assessed through both peripheral and cerebrovascular measures, as the cause, suggesting a direct platelet effect on the neurovascular unit. Clustering analyses further delineated distinct platelet phenotype groups, reinforcing this association. The authors propose that platelets influence neurovascular coupling and may contribute to early neurodegenerative changes.

The strengths of the paper include an interesting integration of platelet functional assays with fMRI measures of neurovascular function; a comprehensive statistical analyses including principal component analysis, clustering, and mediation models; and a control for demographic confounders. The data presentation is excellent and the figures overall are of high quality. The writing is clear.

The authors also do a good job of listing and describing the limitations of this study in the discussion, such as a focus on only visual cortex responses, no behavioural or cognitive relationships explored to platelet activation, and limits of BOLD signal itself in not being able to distinguish between changes in neural activity vs changes in vascular function.

Regarding weaknesses, I have no major problems with this work, but a couple points:

There was a limited exploration of platelet activation markers and potential molecular mechanisms. Can the authors justify their choices versus some others, such as looking at 1) P-selectin, 2) Integrin $\alpha\text{IIb}\beta\text{3}$, or 3) the release of Beta-Thromboglobulin or Platelet Factor 4? This could just be a discussion piece.

Thank you for your insightful suggestion. We appreciate your recommendation to further discuss platelet activation markers and the potential molecular mechanisms by which activated platelets may influence neurovascular function. We have expanded the manuscript to include a discussion on the potential roles of activated platelets and the factors they

release, such as VEGF, TGF- β , PDGF-BB, P-selectin, Beta-Thromboglobulin, and Platelet Factor 4. These factors have well-established roles in modulating vascular tone, blood-brain barrier (BBB) integrity, and neuroinflammation, all of which can impact cerebral circulation and neural function.

Regarding the focus of our study on platelet reactivity, we chose this approach due to practicalities within a complex initial study, where various measures were necessary in a short timeframe after the blood draws (platelet function assays need to be rigorously controlled for time since this will affect responses). We have extensive experience of measuring many of these parameters, and particularly cell markers by flow cytometry. We agree that this would be an important consideration for future targeted studies to explore the underlying mechanisms of these observed relationships. i.e. which feature of platelet activation might explain the impact on neurovascular coupling?

We have now incorporated a more thorough discussion of how activated platelets could impact neurovascular function, including their release of factors such as P-selectin and platelet-derived chemokines, which promote platelet-leukocyte interactions and contribute to endothelial dysfunction. We also explore how serotonin release from activated platelets could modulate vascular tone, and how platelet-derived factors like VEGF, TGF- β , and PDGF-BB may influence BBB permeability and pericyte function over longer timescales. While these mechanisms are speculative in the context of our study, they highlight the important biological overlap between platelet regulation and neurovascular function, which we believe warrants further investigation in future research.

Thank you again for your valuable feedback, and we hope that these additions to the manuscript help clarify our rationale for focusing on platelet reactivity and provide a more comprehensive view of the potential mechanisms linking platelets to neurovascular health.

Discussion added to the manuscript:

In this study, we focused on platelet reactivity assays to indicate platelet function in different physiological contexts. Platelet reactivity provides a more direct measure of platelet responses to specific physiological stimuli, compared to platelet activation markers which primarily reflect static activation status or systemic inflammation. Unlike activation markers, which may be sensitive to acute inflammatory events (e.g., infections, injuries), platelet reactivity assays are less influenced by external stimuli.

While our study focused on platelet reactivity, it is important to consider how activated platelets could impact neurovascular function through various mechanisms. Activated platelets release a range of factors that influence vascular tone, blood-brain barrier (BBB) integrity, and neuroinflammation, which can collectively affect cerebral circulation and neural function. Platelets may influence neurovascular unit by altering vascular integrity. Platelets release vascular endothelial growth factor (VEGF) (Italiano & Shivdasani, 2003), and transforming growth factor-beta (TGF- β) β (Assoian & Sporn, 1986), both of which can directly increase BBB permeability, particularly under hypoxic conditions (Zhang et al., 2000; Schumacher et al., 2023). In addition, TGF- β has been shown to alter the metabolic behaviour of pericytes, shifting their energy production from oxidative phosphorylation to glycolysis (Schumacher et al., 2023). Pericytes play a key role in maintaining the structural and functional integrity of the BBB, as well as regulating capillary flow, angiogenesis, and immune responses (Sá-Pereira et al., 2012; Sweeney et al., 2016; Brown et al.,

2019). Furthermore, platelet-derived growth factor BB (PDGF-BB) signalling promotes pericyte recruitment, proliferation, and migration, contributing to vascular tone regulation and neurovascular function (Gaceb et al., 2018).

Platelet activation is associated with the release of inflammatory mediators that can lead to secondary BBB disruption, immune cell recruitment, and altered cerebral blood flow. The release of P-selectin and chemokines, including Beta-Thromboglobulin (β -TG) and Platelet Factor 4 (PF4), can promote platelet-leukocyte interactions and contribute to endothelial dysfunction (Kutlar & Embury, 2014; Duchene & von Hundelshausen, 2015; Bakogiannis et al., 2019), potentially impairing vascular integrity and blood flow. Interestingly, recent research has shown that administering platelet factor 4 from young mice improved cognition and reduced neuroinflammation in the hippocampus of aged mice (Schroer et al., 2023).

Platelet-derived mediators can activate sensory neurons and contribute to neurogenic inflammation. Platelets express receptors for tachykinins, such as substance P and neurokinin 1 (NK1). These neuropeptides act on neurovascular structures to modulate inflammation and cerebrovascular responses. Research by this group has demonstrated their additional role in platelet thrombus formation and regulation (Graham et al., 2004; Jones et al., 2008). This provides another biological overlap between platelets and the neurovascular unit that extends beyond a systemic contribution to atherosclerosis.

Finally, platelets may affect cerebral neurovascular function through neurotransmitter-mediated vasoregulation. Serotonin release by activated platelets (Mercado & Kilic, 2010) may link platelet activity to neurovascular function by modulating neurotransmission and vascular tone. Serotonin acts on receptors present on neurons, glial cells, and vascular smooth muscle cells (Hoyer et al., 1994) and can induce vasoconstriction, thereby altering blood flow and neural activity (Vanhoutte, 1987). This connection between platelets and vascular tone could influence the regulation of blood flow within the neurovascular unit, impacting both cerebral blood flow and neural function.

These potential mechanisms, while speculative in the context of our study, outline the large degree of biological overlap between platelet regulatory mechanisms and the neurovascular unit. Future research should incorporate both functional reactivity measures and the mechanistic roles of platelet-released factors to further elucidate how platelet reactivity impacts the neurovascular unit and cognitive health.

I would like the authors to refine some of the wording regarding the relationship between platelet factors and NVU function. As the data are primarily correlational, wording like "direct link" or "direct association" should drop the "direct". What is the difference between a direct association and an association? I believe "direct" is unnecessary and potentially misleading/inaccurate. When talking about links or connections, "potential link" should be used instead.

We completely accept your point on this. We have changed our wording throughout the manuscript in line with your suggestion.

We have copied some examples of these changes below

Abstract:

We show an **direct-association** between platelet reactivity and cerebral neurovascular function that is both independent of vascular reactivity and mechanistically specific

Introduction;

We incorporated experimental manipulations that targeted peripheral vascular reactivity and cerebrovascular reactivity to determine whether any association could be attributed to systemic vascular effects, or **direct interactions** between platelets and the neurovascular unit.

Methods:

After initiation of the data analyses, we additionally sought to determine (iv) whether correlations between platelet reactivity and cerebral neurovascular function were **directly** attributable to platelet reactivity *per se*, or simply reflected shared variance driven by demographic differences.

Results (subheading):

Platelet reactivity **directly** affects cerebral neurovascular function, **independent of vascular reactivity**

Discussion:

In conclusion, we show that platelet reactivity was **directly** associated with cerebral neurovascular function in a healthy older sample. This novel **direct** association appears to bypass systemic vascular function, and to be mechanistically specific; distinct platelet signalling mechanisms were associated with different physiological components of the hemodynamic response to neural activity.

END OF COMMENTS

Dear Dr Rossetti,

Re: JP-RP-2025-288405R1 "Evidence for control of cerebral neurovascular function by circulating platelets in healthy older adults" by Gabriella Maria Knasel Rossetti, Joanne L Dunster, Aamir Sohail, Brendan Williams, Kiera M Cox, Suzannah Rawlings, Elysia Jewett, Eleanor Benford, Julie A Lovegrove, Jonathan M Gibbins, and Anastasia Christakou

We are pleased to tell you that your paper has been accepted for publication in The Journal of Physiology.

Yours sincerely,

Harold Schultz
Senior Editor
The Journal of Physiology

If you would like to receive our 'Research Roundup', a monthly newsletter highlighting the cutting-edge research published in The Physiological Society's family of journals (The Journal of Physiology, Experimental Physiology, Physiological Reports, The Journal of Nutritional Physiology and The Journal of Precision Medicine: Health and Disease), please click this link, fill in your name and email address and select 'Research Roundup':
<https://www.physoc.org/journals-and-media/membernews>

- You can help your research get the attention it deserves! Check out Wiley's free Promotion Guide for best-practice recommendations for promoting your work at: www.wileyauthors.com/eeo/guide. You can learn more about Wiley Editing Services which offers professional video, design, and writing services to create shareable video abstracts, infographics, conference posters, lay summaries, and research news stories for your research at: www.wileyauthors.com/eeo/promotion.

EDITOR COMMENTS

Reviewing Editor:

Thank you for addressing the reviewers' comments. I am satisfied with your responses and am pleased to report that your manuscript is considered acceptable for publication in The Journal of Physiology.

Senior Editor:

The editors thank the authors for revision of the manuscript. They are commended on a careful and thorough response to concerns and requests. The article is now accepted for publication. Congratulations for an interesting and insightful study. Please consider the Journal of Physiology for your future studies.